# Double Neural Counterfactual Regret Minimization

## Abstract

Counterfactual Regret Minimization (CRF) is a fundamental and effective technique for solving Imperfect Information Games (IIG). However, the original CRF algorithm only works for discrete state and action spaces, and the resulting strategy is maintained as a tabular representation. Such tabular representation limits the method from being directly applied to large games and continuing to improve from a poor strategy profile. In this paper, we propose a double neural representation for the imperfect information games, where one neural network represents the cumulative regret, and the other represents the average strategy. Furthermore, we adopt the counterfactual regret minimization algorithm to optimize this double neural representation. To make neural learning efficient, we also developed several novel techniques including a robust sampling method, mini-batch Monte Carlo Counterfactual Regret Minimization (MCCFR) and Monte Carlo Counterfactual Regret Minimization Plus (MCCFR+) which may be of independent interests. Experimentally, we demonstrate that the proposed double neural algorithm converges significantly better than the reinforcement learning counterpart.

## 1 Introduction

In Imperfect Information Games (**IIG**), a player only has partial access to the knowledge of her opponents before making a decision. This is similar to real-world scenarios, such as trading, traffic routing, and public auction. Thus designing methods for solving IIG is of great economic and societal benefits. Due to the hidden information, a player has to reason under the uncertainty about her opponents' information, and she also needs to act so as to take advantage of her opponents' uncertainty about her own information.

Nash equilibrium is a typical solution concept for a two-player extensive-form game. Many algorithms have been designed over years to approximately find Nash equilibrium for large games. One of the most effective approaches is CFR (Zinkevich et al., 2007). In this algorithm, the authors proposed to minimize overall counterfactual regret and prove that the average of the strategies in all iterations would converge to a Nash equilibrium. However, the original CFR only works for discrete state and action spaces, and the resulting strategy is maintained as a tabular representation. Such tabular representation limits the method from being directly applied to large games and continuing to improve if starting from a poor strategy profile.

To alleviate CFR's large memory requirement in large games such as heads-up no-limit Texas Hold'em, Moravcik et al. (2017) proposed a seminal approach called DeepStack which uses fully connected neural networks to represent players counterfactual values and obtain a strategy online as requested. However, the strategy is still represented as a tabular form and the quality of this solution depends a lot on the initial quality of the counterfactual network. Furthermore, the counterfactual network is estimated separately, and it is not easy to continue improving both counterfactual network and the tabular strategy profile in an end-to-end optimization framework.

Heinrich et al. (2015); Heinrich & Silver (2016) proposed end-to-end fictitious self-play approaches (XFP and NFSP respectively) to learn the approximate Nash equilibrium with deep reinforcement learning. In a fictitious play model, strategies are represented as neural networks and the strategies are updated by selecting the best responses to their opponents' average strategies. This approach is advantageous in the sense that the approach does not rely on abstracting the game, and in theory, the strategy should continually improve as the algorithm iterates more steps. However, these methods

do not explicitly take into account the hidden information in a game, because they are optimized based on the transition memory and the reward of the intermediate node is the utility of the game rather than the counterfactual value which consider the distribution of hidden variables (opponent's private information). In experiments for games such as Leduc Hold'em, these methods converge slower than tabular based counterfactual regret minimization algorithms. Waugh et al. (2015) used handcraft features of the information sets to estimates the counterfactual regret. However, it need traverse the full game tree which is infeasible in large games.

Thus it remains an open question whether the purely neural-based end-to-end approach can achieve comparable performance to tabular based CFR approach. In the paper, we partially resolve this open question by designing a double neural counterfactual regret minimization algorithm which can match the performance of tabular based counterfactual regret minimization algorithm. We employed two neural networks, one for the cumulative regret, and the other for the average strategy. We show that careful algorithm design allows these two networks to track the cumulative regret and average strategy respectively, resulting in a converging neural strategy. Furthermore, in order to improve the convergence of the neural algorithm, we also developed a new sampling technique which has lower variance than the outcome sampling, while being more memory efficient than the external sampling. In experiments with One-card poker and a large Leduc Hold'em containing more than $10^7$ nodes, we showed that the proposed double neural algorithm has a strong generalization and compression ability even though only a small proportion of nodes are visited in each iteration. In addition, this method can converge to comparable results produced by its tabular counterpart while performing much better than deep reinforcement learning method. The current results open up the possibility for a purely neural approach to directly solve large IIG.

## 2 BACKGROUND

In this section, we will introduce some background on IIG and existing approaches to solve them.

### 2.1 REPRESENTATION OF EXTENSIVE-FORM GAME

We define the components of an extensive-form game following Osborne & Ariel (1994) (page $200 \sim 201$). A finite set $N = \{0, 1, ..., n-1\}$ of **players**. Define $h_i^v$ as the **hidden variable** of player $i$ in IIG, $e.g.$, in poker game $h_i^v$ refers to the private cards of player $i$. $H$ refers to a finite set of histories. Each member $h = (h_i^v)_{i=0,1,...,n-1}(a_l)_{l=0,...,L-1} = h_0^v h_1^v ... h_{n-1}^v a_0 a_1 ... a_{L-1}$ of $H$ denotes a possible **history** (or state), which consists of each player's hidden variable and $L$ actions taken by players including chance. For player $i$, $h$ also can be denoted as $h_i^v h_{-i}^v a_0 a_1 ... a_{L-1}$, where $h_{-i}^v$ refers to the opponent's hidden variables. The empty sequence $\emptyset$ is a member of $H$. $h_j \sqsubseteq h$ denotes $h_j$ is a prefix of $h$, where $h_j = (h_i^v)_{i=0,1,...,n-1}(a_l)_{l=1,...,L'-1}$ and $0 < L' < L$. $Z \subseteq H$ denotes the terminal histories and any member $z \in Z$ is not a prefix of any other sequences. $A(h) = \{a : ha \in H\}$ is the set of available actions after non-terminal history $h \in H \setminus Z$. A **player function** $P$ assigns a member of $N \cup \{c\}$ to each non-terminal history, where $c$ denotes the chance player id, which usually is -1. $P(h)$ is the player who takes an action after history $h$. $\mathcal{I}_i$ of a history $\{h \in H : P(h) = i\}$ is an **information partition** of player $i$. A set $I_i \in \mathcal{I}_i$ is an **information set** of player $i$ and $I_i(h)$ refers to information set $I_i$ at state $h$. Generally, $I_i$ could only remember the information observed by player $i$ including player $i's$ hidden variable and public actions. Therefore $I_i$ indicates a sequence in IIG, $i.e.$, $h_i^v a_0 a_2 ... a_{L-1}$. For $I_i \in \mathcal{I}_i$ we denote by $A(I_i)$ the set $A(h)$ and by $P(I_i)$ the player $P(h)$ for any $h \in I_i$. For each player $i \in N$ a utility function $u_i(z)$ define the payoff of the terminal state $z$. A more detailed explanation of these notations and definitions is presented in section B.

### 2.2 STRATEGY AND NASH EQUILIBRIUM

A **strategy profile** $\sigma = \{\sigma_i | \sigma_i \in \Sigma_i, i \in N\}$ is a collection of strategies for all players, where $\Sigma_i$ is the set of all possible strategies for player $i$. $\sigma_{-i}$ refers to strategy of all players other than player $i$. For play $i \in N$ the **strategy** $\sigma_i(I_i)$ is a function, which assigns an action distribution over $A(I_i)$ to information set $I_i$. $\sigma_i(a|h)$ denotes the probability of action $a$ taken by player $i \in N \cup \{c\}$ at state $h$. In IIG, $\forall h_1, h_2 \in I_i$, we have $I_i = I_i(h_1) = I_i(h_2)$, $\sigma_i(I_i) = \sigma_i(h_1) = \sigma_i(h_2)$, $\sigma_i(a|I_i) = \sigma_i(a|h_1) = \sigma_i(a|h_2)$. For iterative method such as CFR, $\sigma^t$ refers to the strategy profile at $t$-th iteration. The **state reach probability** of history $h$ is denoted by $\pi^\sigma(h)$ if players take actions according to $\sigma$. For an empty sequence $\pi^\sigma(\emptyset) = 1$. The reach probability can be decomposed into $\pi^\sigma(h) = \prod_{i \in N \cup \{c\}} \pi_i^\sigma(h) = \pi_i^\sigma(h) \pi_{-i}^\sigma(h)$ according to each player's contribution, where $\pi_i^\sigma(h) = \prod_{h'a \sqsubseteq h, P(h') = P(h)} \sigma_i(a|h')$ and $\pi_{-i}^\sigma(h) = \prod_{h'a \sqsubseteq h, P(h') \neq P(h)} \sigma_{-i}(a|h')$. The **information set reach probability** of $I_i$ is defined as $\pi^\sigma(I_i) = \sum_{h \in I_i} \pi^\sigma(h)$. If $h' \sqsubseteq h$, the **interval state reach**

**probability** from state $h'$ to $h$ is defined as $\pi^\sigma(h', h)$, then we have $\pi^\sigma(h', h) = \pi^\sigma(h)/\pi^\sigma(h')$. $\pi_i^\sigma(I_i)$, $\pi_{-i}^\sigma(I_i)$, $\pi_i^\sigma(h', h)$, and $\pi_{-i}^\sigma(h', h)$ are defined similarly.

## 2.3 COUNTERFACTUAL REGRET MINIMIZATION

In large and zero-sum IIG, CFR is proved to be an efficient method to compute Nash equilibrium (Zinkevich et al., 2007; Brown & Sandholm, 2017). We present some key ideas of this method as follows.

**Lemma 1:** The state reach probability of one player is proportional to posterior probability of the opponent's hidden variable, *i.e.*, $p(h_{-i}^v|I_i) \propto \pi_{-i}^\sigma(h)$, where $h_i^v$ and $I_i$ indicate a particular $h$. (see the proof in section G.1)

For player $i$ and strategy profile $\sigma$, the **counterfactual value (CFV)** $v_i^\sigma(h)$ at state $h$ is define as

$$v_i^\sigma(h) = \sum_{h \sqsubseteq z, z \in Z} \pi_{-i}^\sigma(h)\pi^\sigma(h, z)u_i(z) = \sum_{h \sqsubseteq z, z \in Z} \pi_i^\sigma(h, z)u_i'(z). \tag{1}$$

where $u_i'(z) = \pi_{-i}^\sigma(z)u_i(z)$ is the expected reward of player $i$ with respective to the approximated posterior distribution of the opponent's hidden variable. The **action counterfactual value** of taking action $a$ is $v_i^\sigma(a|h) = v_i^\sigma(ha)$ and the regret of taking this action is $r_i^\sigma(a|h) = v_i^\sigma(a|h) - v_i^\sigma(h)$. Similarly, the CFV of information set $I_i$ is $v_i^\sigma(I_i) = \sum_{h \in I_i} v_i^\sigma(h)$ and the regret is $r_i^\sigma(a|I_i) = \sum_{z \in Z, ha \sqsubseteq z, h \in I_i} \pi_i^\sigma(ha, z)u_i'(z) - \sum_{z \in Z, h \sqsubseteq z, h \in I_i} \pi_i^\sigma(h, z)u_i'(z)$. Then the **cumulative regret** of action $a$ after $T$ iterations is

$$R_i^T(a|I_i) = \sum_{t=1}^T (v_i^{\sigma^t}(a|I_i) - v_i^{\sigma^t}(I_i)) = R_i^{T-1}(a|I_i) + r_i^{\sigma^T}(a|I_i). \tag{2}$$

where $R_i^0(a|I_i) = 0$. Define $R_i^{T,+}(a|I_i) = \max(R_i^T(a|I_i), 0)$, the **current strategy (or behavior strategy)** at $T+1$ iteration will be updated by

$$\sigma_i^{T+1}(a|I_i) = \begin{cases} \frac{R_i^{T,+}(a|I_i)}{\sum_{a \in A(I_i)} R_i^{T,+}(a|I_i)} & \text{if } \sum_{a \in A(I_i)} R_i^{T,+}(a|I_i) > 0 \\ \frac{1}{|A(I_i)|} & \text{otherwise.} \end{cases} \tag{3}$$

The **average strategy** $\bar{\sigma}_i^T$ from iteration 1 to $T$ is defined as:

$$\bar{\sigma}_i^T(a|I_i) = \frac{\sum_{t=1}^T \pi_i^{\sigma^t}(I_i)\sigma_i^t(a|I_i)}{\sum_{t=1}^T \pi_i^{\sigma^t}(I_i)}. \tag{4}$$

where $\pi_i^{\sigma^t}(I_i)$ denotes the information set reach probability of $I_i$ at $t$-th iteration and is used to weight the corresponding current strategy $\sigma_i^t(a|I_i)$. Define $s_i^t(a|I_i) = \pi_i^{\sigma^t}(I_i)\sigma_i^t(a|I_i)$ as the additional numerator in iteration $t$, then the cumulative numerator can be defined as

$$S^T(a|I_i) = \sum_{t=1}^T \pi_i^{\sigma^t}(I_i)\sigma_i^t(a|I_i) = S^{T-1}(a|I_i) + s_i^T(a|I_i). \tag{5}$$

where $S^0(a|I_i) = 0$.

## 2.4 MONTE CARLO CFR

When solving a game, CFR needs to traverse the entire game tree in each iteration, which will prevent it from handling large games with limited memory. To address this challenge, Lanctot et al. (2009) proposed a Monte Carlo CFR to minimize counterfactual regret. Their method can compute an unbiased estimation of counterfactual value and avoid traversing the entire game tree. Since only subsets of all information sets are visited in each iteration, this approach requires less memory than standard CFR.

Define $\mathcal{Q} = \{Q_1, Q_2, ..., Q_m\}$, where $Q_j \in Z$ is a **block** of sampling terminal histories in each iteration, such that $\mathcal{Q}_j$ spans the set $Z$. Generally, different $Q_j$ may have an overlap according to the specify sampling schema. Specifically, in the external sampling and outcome sampling, each block $Q_j \in \mathcal{Q}$ is a partition of $Z$. Define $q_{Q_j}$ as the probability of considering block $Q_j$, where $\sum_{j=1}^m q_{Q_j} = 1$. Define $q(z) = \sum_{j:z \in Q_j} q_{Q_j}$ as the probability of considering a particular terminal history $z$. Specifically, vanilla CFR is a special case of MCCFR, where $\mathcal{Q} = \{Z\}$ only contain one block and $q_{Q_1} = 1$. In outcome sampling, only one trajectory will be sampled, such that $\forall Q_j \in \mathcal{Q}$,

$|Q_j| = 1$ and $|\mathcal{Q}_j| = |Z|$. For information set $I_i$, a sample estimate of **counterfactual value** is $\tilde{v}_i^\sigma(I_i|Q_j) = \sum_{h \in I_i, z \in Q_j, h \sqsubseteq z} \frac{1}{q(z)} \pi_{-i}^\sigma(z) \pi_i^\sigma(h,z) u_i(z)$.

**Lemma 2:** The sampling counterfactual value in MCCFR is the unbiased estimation of actual counterfactual value in CFR. $E_{j \sim q_{Q_j}}[\tilde{v}_i^\sigma(I_i|Q_j)] = v_i^\sigma(I_i)$ (Lemma 1, Lanctot et al. (2009).)

Define $\sigma^{rs}$ as **sampling strategy profile**, where $\sigma_i^{rs}$ is the sampling strategy for player $i$ and $\sigma_{-i}^{rs}$ are the sampling strategies for players expect $i$. Particularly, for both external sampling and outcome sampling proposed by (Lanctot et al., 2009), $\sigma_{-i}^{rs} = \sigma_{-i}$. The regret of the sampled action $a \in A(I_i)$ is defined as

$$\tilde{r}_i^\sigma((a|I_i)|Q_j) = \sum_{z \in Q_j, ha \sqsubseteq z, h \in I_i} \pi_i^\sigma(ha, z) u_i^{rs}(z) - \sum_{z \in Q_j, h \sqsubseteq z, h \in I_i} \pi_i^\sigma(h, z) u_i^{rs}(z) \quad , \quad (6)$$

where $u_i^{rs}(z) = \frac{u_i(z)}{\pi_i^{\sigma^{rs}}(z)}$ is a new utility weighted by $\frac{1}{\pi_i^{\sigma^{rs}}(z)}$. The sample estimate for **cumulative regret** of action $a$ after $T$ iterations is $\tilde{R}_i^T((a|I_i)|Q_j) = \tilde{R}_i^{T-1}((a|I_i)|Q_j) + \tilde{r}_i^{\sigma^T}((a|I_i)|Q_j)$ with $\tilde{R}_i^0((a|I_i)|Q_j) = 0$.

# 3 DOUBLE NEURAL COUNTERFACTUAL REGRET MINIMIZATION

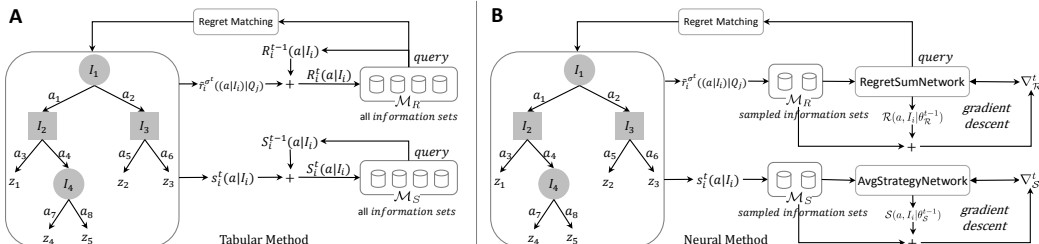

Figure 1: (A) tabular based CRF and (B) our double neural based CRF framework.

In this section, we will explain our double neural CFR algorithm, where we employ two neural networks, one for the cumulative regret, and the other for the average strategy. As shown in Figure 1 (A), standard CFR-family methods such as CFR (Zinkevich et al., 2007), outcome-sampling MCCFR, external sampling MCCFR (Lanctot et al., 2009), and CFR+ (Tammelin, 2014) need to use two large tabular-based memories $\mathcal{M}_R$ and $\mathcal{M}_S$ to record the cumulative regret and average strategy for all information sets. Such tabular representation makes these methods difficult to apply to large extensive-form games with limited time and space (Burch, 2017).

In contrast, we will use two deep neural networks to compute approximate Nash equilibrium of IIG as shown in Figure 1 (B). Different from NFSP, our method is based on the theory of CFR, where the first network is used to learn the cumulative regret and the other is to learn the cumulative numerator of the average strategy profile. With the help of these two networks, we do not need to use two large tabular-based memories; instead, we rely on the generalization ability of the compact neural network to produce the cumulative regret and the average strategy. In practice, the proposed double neural method can achieve a lower exploitability with fewer iterations than NFSP. In addition, we present experimentally that our double neural CFR can also continually improve after initialization from a poor tabular strategy.

## 3.1 OVERALL FRAMEWORK

The iterative updates of the CFR algorithm maintain two strategies: the current strategy $\sigma_i^t(a|I_i)$, and the average strategy $\bar{\sigma}_i^t(a|I_i)$ for $\forall i \in N, \forall I_i \in \mathcal{I}_i, \forall a \in A(I_i), \forall t \in \{1, \ldots, T\}$. Thus, our two neural networks are designed to maintain these two strategies in iterative fashion. More specifically,

- **Current strategy.** According to Eq. (3), current strategy $\sigma^{t+1}(a|I_i)$ is computed by the cumulative regret $R^t(a|I_i)$. We only need to track the numerator in Eq. (3) since the normalization in the denominator can easily be computed when the strategy is used. Given information set $I_i$ and action $a$, we design a neural network **RegretSumNetwork(RSN)** $\mathcal{R}(a, I_i|\theta_\mathcal{R}^t)$ to learn $R^t(a|I_i)$, where $\theta_\mathcal{R}^t$ is the parameter in the network at $t$-th iteration. As shown Figure 1 (b), define memory $\mathcal{M}_R = \{(I_i, \tilde{r}_i^{\sigma^t}((a|I_i)|Q_j))|\forall i \in N, \forall a \in A(I_i), h \in I_i, h \sqsubseteq z, z \in Q_j\}$. Each member of $\mathcal{M}_R$ is the visited information set $I_i$ and

the corresponding regret $\tilde{r}_i^{\sigma^t}((a|I_i)|Q_j)$, where $Q_j$ is the sampled block in $t$-th iteration. According to Eq. (2), we can estimate $\mathcal{R}(a, I_i|\theta_{\mathcal{R}}^{t+1})$ using the following optimization:

$$\theta_{\mathcal{R}}^{t+1} \leftarrow \underset{\theta_{\mathcal{R}}^{t+1}}{\operatorname{argmin}} \sum_{(I_i, \tilde{r}_i^{\sigma^t}((a|I_i)|Q_j)) \in \mathcal{M}_R} \left( \mathcal{R}(a, I_i|\theta_{\mathcal{R}}^t) + \tilde{r}_i^{\sigma^t}((a|I_i)|Q_j) - \mathcal{R}(a, I_i|\theta_{\mathcal{R}}^{t+1}) \right)^2. \quad (7)$$

- **Average Strategy.** According to Eq. (4), the approximate Nash equilibrium is the weighted average of all previous strategies over $T$ iterations. Similar to the cumulative regret, we employ another deep neural network **AvgStrategyNetwork(ASN)** to learn the numerator of the average strategy. Define $\mathcal{M}_S = \{(I_i, \pi_i^{\sigma^t}(I_i)\sigma_i^t(a|I_i))|\forall i \in N, \forall a \in A(I_i), h \in I_i, h \sqsubseteq z, z \in Q_j\}$. Each member of $\mathcal{M}_S$ is the visited information set $I_i$ and the value of $\pi_i^{\sigma^t}(I_i)\sigma_i^t(a|I_i)$, where $Q_j$ is the sampled block in $t$-th iteration. Then the parameter $\theta_{\mathcal{S}}^{t+1}$ can estimated by the following optimization:

$$\theta_{\mathcal{S}}^{t+1} \leftarrow \underset{\theta_{\mathcal{S}}^{t+1}}{\operatorname{argmin}} \sum_{(I_i, s_i^t(a|I_i)) \in \mathcal{M}_S} \left( \mathcal{S}(a, I_i|\theta_{\mathcal{S}}^t) + s_i^t(a|I_i) - \mathcal{S}(a, I_i|\theta_{\mathcal{S}}^{t+1}) \right)^2. \quad (8)$$

**Remark 1:** In each iteration, only a small subset of information sets are sampled, which may lead to the neural networks forgetting values for those unobserved information sets. To address this problem, we will use the neural network parameters from the previous iteration as the initialization, which gives an online learning/adaptation flavor to the updates. Furthermore, due to the generalization ability of the neural networks, even samples from a small number of information sets are used to update the new neural networks, the newly updated neural networks can produce very good value for the cumulative regret and the average strategy.

**Remark 2:** As we increase the number of iterations $t$, the value of $R_i^t(a|I_i)$ will become increasingly large, which may make neural network difficult to learn. To address this problem, we will normalize the cumulative regret by a factor of $\sqrt{t}$ to make its range more stable. This can be understood from the regret bound of online learning. More specifically, let $\Delta = \max_{I_i, a, t} |R^t(a|I_i) - R^{t-1}(a|I_i)|, \forall I_i \in \mathcal{I}, a \in A(I_i), t \in \{1, \ldots, T\}$. We have $R_i^t(a|I_i) \leq \Delta\sqrt{|A|t}$ according to the Theorem 6 in (Burch, 2017), where $|A| = \max_{I_i \in \mathcal{I}} |A(I_i)|$. In practice, we can use the neural network to track $\hat{R}_i^t(a|I_i) = R_i^t(a|I_i)/\sqrt{t}$, and update it by

$$\hat{R}_i^t(a|I_i) = \frac{\sqrt{t-1}\hat{R}_i^{t-1}(a|I_i)}{\sqrt{t}} + \frac{r_i^{\sigma^t}(a|I_i)}{\sqrt{t}}, \text{ where } \hat{R}_i^0(a|I_i) = 0. \quad (9)$$

**Remark 3:** The optimization problem for the double neural networks is different from that in DQN (Mnih et al., 2015). In DQN, the Q-value for the greedy action is used in the update, while in our setting, we do not use greedy actions. Algorithm E gives further details on how to optimize the objectives in Eq. (7) and Eq. (8).

**Relation between CFR, MCCFR and our double neural method**. As shown in Figure 1, these three methods are based on the CFR framework. The CFR computes counterfactual value and regret by traversing the entire tree in each iteration, which makes it computationally intensive to be applied to large games directly. MCCFR samples a subset of information sets and will need less computation than CFR in each iteration. However, both CFR and MCCFR need two large tabular memories to save the cumulative regrets and the numerators of the average strategy for all information sets, which prevents these two methods to be used in large games directly. The proposed neural method keeps the benefit of MCCFR yet without the need for two large tabular memories.

## 3.2 RECURRENT NEURAL NETWORK REPRESENTATION FOR INFORMATION SET

In order to define our $\mathcal{R}$ and $\mathcal{S}$ network, we need to represent the information set $I_i \in \mathcal{I}$ in extensive-form games. In such games, players take action in alternating fashion and each player makes a decision according to the observed history. Because the action sequences vary in length, in this paper, we model them with a recurrent neural network and each action in the sequence corresponds to a cell in RNN. This architecture is different from the one in DeepStack (Moravcik et al., 2017), which used a fully connected deep neural network to estimate counterfactual value. Figure 2 (A) provides an illustration of the proposed deep sequential neural network representation for information sets. Besides the vanilla RNN, there are several variants of more expressive RNNs,

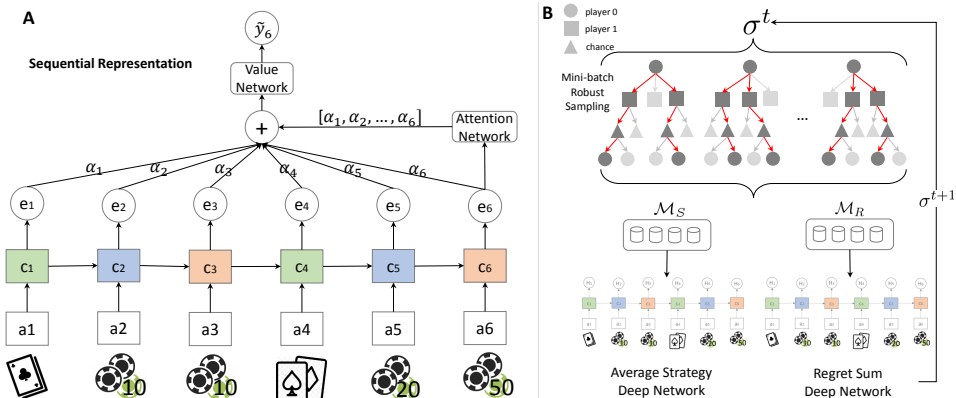

Figure 2: (A) the key architecture of the sequential neural networks. (B) an overview of the novel double neural counterfactual regret minimization method.

such as the GRU (Cho et al., 2014) and LSTM (Hochreiter & Schmidhuber, 1997). In our later experiments, we will compare these different neural architectures as well as a fully connected network representation.

Furthermore, different position in the sequence may contribute differently to the decision making, we will add an attention mechanism (Desimone & Duncan, 1995; Cho et al., 2015) to the RNN architecture to enhance the representation. For example, the player may need to take a more aggressive strategy after beneficial public cards are revealed. Thus the information, after the public cards are revealed may be more important. In practice, we find that the attention mechanism can help the double neural CFR obtain a better convergence rate. In section D, we will provide more details on neural network architectures.

### 3.3 CONTINUAL IMPROVEMENT

With the proposed framework of double neural CFR, it is easy to initialize the neural networks from an existing strategy profile based on the tabular representation or neural representation. For information set $I_i$ and action $a$, in an existing strategy profile, define $R'_i(a|I_i)$ as the cumulative regret and $S'(a|I_i)$ as the cumulative numerator of average strategy. We can clone the cumulative regret for all information sets and actions by optimizing

$$\theta_{\mathcal{R}}^* \leftarrow \operatorname*{argmin}_{\theta_{\mathcal{R}}} \sum_{i \in N, I_i \in \mathcal{I}_i, a \in A(I_i)} \left( \mathcal{R}(a, I_i | \theta_{\mathcal{R}}) - R'(a|I_i) \right)^2. \tag{10}$$

Similarly, the parameters $\theta_{\mathcal{S}}^*$ for cloning the cumulative numerator of average strategy can be optimized in the same way. Based on the learned $\theta_{\mathcal{R}}^*$ and $\theta_{\mathcal{S}}^*$, we can warm start the double neural networks and continually improve beyond the tabular strategy profile.

**Remark:** In the large extensive game, the initial strategy is obtained from an abstracted game which has a manageable number of information sets. The abstracted game is generated by domain knowledge, such as clustering similar hand strength cards into the same buckets. Once the strategy of this abstract game is solved, it can be clone according to Eq. (10) and improved continuously using our double neural CFR framework.

### 3.4 OVERALL ALGORITHM

Algorithm 1 provides a summary of the proposed double neural counterfactual regret minimization algorithm. In the first iteration, if the system warm starts from tabular based CFR or MCCFR methods, the techniques in section 3.3 will be used to clone the cumulative regrets and strategy. If there is no warm start initialization, we can start our algorithm by randomly initializing the parameters in RSN and ASN at iteration $t = 1$. Then sampling methods will return the counterfactual regret and the numerator of average strategy for the sampled information sets in this iteration, and they will be saved in memories $\mathcal{M}_{\mathcal{R}}$ and $\mathcal{M}_{\mathcal{S}}$ respectively. Then these samples will be

used by the NeuralAgent algorithm from Algorithm 2 to optimize RSN and ASN. Further details for the sampling methods and the NeuralAgent fitting algorithm will be discussed in the next section.

---

**Algorithm 1:** Counterfactual Regret Minimization with Two Deep Neural Networks

1 **Function** `Agent` $(T, b)$ **:**
2    **For** $t = 1$ *to* $T$ **do**
3      **if** $t = 1$ **and** *using warm starting* **then**
4        initialize $\theta_{\mathcal{R}}^t$ and $\theta_{\mathcal{S}}^t$ from an existing checkpoint
5        $t \leftarrow t + 1$             ▷ skip cold starting
6      **else**
7        initialize $\theta_{\mathcal{R}}^t$ and $\theta_{\mathcal{S}}^t$ randomly.
8      $\mathcal{M}_{\mathcal{R}}, \mathcal{M}_{\mathcal{S}} \leftarrow$ sampling methods for CFV and average strategy.      ▷ such as Algorithm3
9      sum aggregate the value in $\mathcal{M}_R$ by information set.      ▷ according to the **Lemma 5** and Equation 12
10      remove duplicated records in $\mathcal{M}_S$.
11      $\theta_{\mathcal{R}}^t \leftarrow \text{NeuralAgent}(\mathcal{R}(\cdot|\theta_{\mathcal{R}}^{t-1}), \mathcal{M}_R, \theta_{\mathcal{R}}^{t-1}, \beta_{\mathcal{R}}^*)$      ▷ update $\theta_{\mathcal{R}}^t$ using Algorithm2
12      $\theta_{\mathcal{S}}^t \leftarrow \text{NeuralAgent}(\mathcal{S}(\cdot|\theta_{\mathcal{S}}^{t-1}), \mathcal{M}_S, \theta_{\mathcal{S}}^{t-1}, \beta_{\mathcal{S}}^*)$      ▷ update $\theta_{\mathcal{S}}^t$ using Algorithm2
13    **return** $\theta_{\mathcal{R}}^t, \theta_{\mathcal{S}}^t$

---

## 4 EFFICIENT TRAINING

In this section, we will propose two techniques to improve the efficiency of the double neural method. These techniques can also be used separately in other CFR-based methods.

### 4.1 ROBUST SAMPLING TECHNIQUES

In this paper, we proposed a new robust sampling technique which has lower variance than outcome sampling, while being more memory efficient than the external sampling. In this robust sampling method, the sampling profile is defined as $\sigma^{rs(k)} = (\sigma_i^{rs(k)}, \sigma_{-i})$, where player $i$ will randomly select $k$ actions according to sampling strategy $\sigma_i^{rs(k)}(I_i)$ for each information set $I_i$ and other players will randomly select one action according to strategy $\sigma_{-i}$.

Specifically, if player $i$ randomly selects $min(k, |A(I_i)|)$ actions according to discrete uniform distribution $unif(0, |A(I_i)|)$ at information set $I_i$, i.e., $\sigma_i^{rs(k)}(a|I_i) = \frac{min(k,|A(I_i)|)}{|A(I_i)|}$, then

$$\pi_i^{\sigma^{rs(k)}}(I_i) = \prod_{h \in I_i, h' \sqsubseteq h, h'a \sqsubseteq h, h' \in I_i'} \frac{min(k, |A(I_i')|)}{|A(I_i')|} \tag{11}$$

and the weighted utility $u_i^{rs(k)}(z)$ will be a constant number in each iteration, which has a low variance. In addition, because the weighted utility no longer requires explicit knowledge of the opponent's strategy, we can use this sampling method for online regret minimization. For simplicity, $k = max$ refers to $k = max_{I_i \in \mathcal{I}} |A(I_i)|$ in the following sections.

**Lemma 3:** If $k = max$ and $\forall i \in N, \forall I_i \in \mathcal{I}_i, \forall a \in A(I_i), \sigma_i^{rs(k)}(a|I_i) \sim unif(0, |A(I_i)|)$, then robust sampling is the same as external sampling.

**Lemma 4:** If $k = 1$ and $\sigma_i^{rs(k)} = \sigma_i$, then robust sampling is the same as outcome sampling.

**Lemma** 3 and **Lemma** 4 provide the relationship between outcome sampling, external sampling, and the proposed robust sampling algorithm. The detailed theoretical analysis are presented in Appendix G.2.

### 4.2 MINI-BATCH TECHNIQUES

**Mini-batch MCCFR:** Traditional outcome sampling and external sampling only sample one block in an iteration and provide an unbiased estimator of origin CFV according to **Lemma 2**. In this paper, we present a mini-batch Monte Carlo technique and randomly sample $b$ blocks in one iteration. Let $Q^j$ denote a block of terminals sampled according to the scheme in section 4.1 at $j-$th time, then **mini-batch CFV** with $b$ mini-batches for information set $I_i$ can be defined as

$$\tilde{v}_i^\sigma(I_i|b) = \frac{1}{b} \sum_{j=1}^{b} \left( \sum_{h \in I_i, z \in Q^j, h \sqsubseteq z} \frac{\pi_{-i}^\sigma(z) \pi_i^\sigma(h, z) u_i(z)}{q(z)} \right) = \sum_{j=1}^{b} \frac{\tilde{v}_i^\sigma(I_i|Q^j)}{b}. \tag{12}$$

Furthermore, we can show that $\tilde{v}_i^\sigma(I_i|b)$ is an unbiased estimator of the counterfactual value of $I_i$:
**Lemma 5:** $E_{Q^j \sim \text{Robust Sampling}}[\tilde{v}_i^\sigma(I_i|b)] = v_i^\sigma(I_i)$. (see the proof in section G.3) Similarly, the **cumulative mini-batch regret** of action $a$ is

$$\tilde{R}_i^T((a|I_i)|b) = \tilde{R}_i^{T-1}((a|I_i)|b) + \tilde{v}_i^{\sigma^T}((a|I_i)|b) - \tilde{v}_i^{\sigma^T}(I_i|b) \qquad , \qquad (13)$$

where $\tilde{R}_i^0((a|I_i)|b) = 0$. In practice, mini-batch technique can sample $b$ blocks in parallel and help MCCFR to converge faster.

**Mini-Batch MCCFR+:** When optimizing counterfactual regret, CFR+ (Tammelin, 2014) substitutes the regret-matching algorithm (Hart & Mas-Colell, 2000) with regret-matching+ and can converge faster than CFR. However, Burch (2017) showed that MCCFR+ actually converge slower than MCCFR when mini-batch is not used. In our paper, we derive mini-batch version of MCCFR+ which updates cumulative mini-batch regret $\tilde{R}^{T,+}((a|I_i)|b)$ up to iteration $T$ by

$$\tilde{R}^{T,+}((a|I_i)|b) = \begin{cases} \left(\tilde{v}_i^{\sigma^T}((a|I_i)|b) - \tilde{v}_i^{\sigma^T}(I_i|b)\right)^+ & \text{if } T = 0 \\ \left(\tilde{R}_i^{T-1,+}((a|I_i)|b) + \tilde{v}_i^{\sigma^T}((a|I_i)|b) - \tilde{v}_i^{\sigma^T}(I_i|b)\right)^+ & \text{if } T > 0 \end{cases}, \qquad (14)$$

where $(x)^+ = max(x,0)$. In practice, we find that mini-batch MCCFR+ converges faster than mini-batch MCCFR when specifying a suitable mini-batch size.

## 5 EXPERIMENT

The proposed double neural CFR algorithm will be evaluated in the One-Card-Poker game with 5 cards and a large No-Limit Leduc Hold'em (NLLH) with stack size 5, 10, and 15. The largest NLLH in our experiment has over $2 \times 10^7$ states and $3.7 \times 10^6$ information sets. We will compare it with tabular CFR and deep reinforcement learning based method such as NFSP. The experiments show that the proposed double neural algorithm can converge to comparable results produced by its tabular counterpart while performing much better than deep reinforcement learning method. With the help of neural networks, our method has a strong generalization ability to converge to an approximate Nash equilibrium by using fewer parameters than the number of information sets. The current results open up the possibility for a purely neural approach to directly solve large IIG. Due to space limit, we present experimental results for One-Card-Poker and the analysis in section C. The hyperparameters and setting about the neural networks can be found in section E.

**Settings.** To simplify the expression, the abbreviations of different methods are defined as follows. **XFP** refers to the full-width extensive-form fictitious play method. **NFSP** refers to the reinforcement learning based fictitious self-play method. **RS-MCCFR** refers to the proposed robust sampling MCCFR. This method with regret matching+ acceleration technique is denoted by **RS-MCCFR+**. These methods only containing one neural network are denoted by **RS-MCCFR+-RSN** and **RS-MCCFR+-ASN** respectively. **RS-MCCFR+-RSN-ASN** refers to the proposed double neural MCCFR. According to **Lemma 3**, if $k = max$, ES-MCCFR is the same with RS-MCCFR. More specifically, we investigated the following questions.

**Is mini-batch sampling helpful?** Figure 3(A) presents the convergence curves of the proposed robust sampling method with $k = max$ under different mini-batch sizes (b=1, 1000, 5000, 10000 respectively). The experimental results show that larger batch sizes generally lead to better strategy profiles. Furthermore, the convergence for $b = 5000$ is as good as $b = 10000$. Thus in the later experiments, we set the mini-batch size equal to 5000.

**Is robust sampling helpful?** Figure 3 (B) and (C) presents convergence curves for outcome sampling, external sampling($k = max$) and the proposed robust sampling method under the different number of sampled actions. The outcome sampling cannot converge to a low exploitability smaller than 0.1 after 1000 iterations The proposed robust sampling algorithm with $k = 1$, which only samples one trajectory like the outcome sampling, can achieve a better strategy profile after the same number of iterations. With an increasing $k$, the robust sampling method achieves an even better convergence rate. Experiment results show $k = 3$ and $5$ have a similar trend with $k = max$, which demonstrates that the proposed robust sampling achieves similar strategy profile but requires less memory than the external sampling. We choose $k = 3$ for the later experiments in Leduc Hold'em Poker. Figure 3 (C) presents the results in a different way and displays the relation between exploitability and the cumulative number of touched nodes. The robust sampling with small $k$ is just as good as the external sampling while being more memory efficient on the condition that each algorithm touches the same number of nodes.

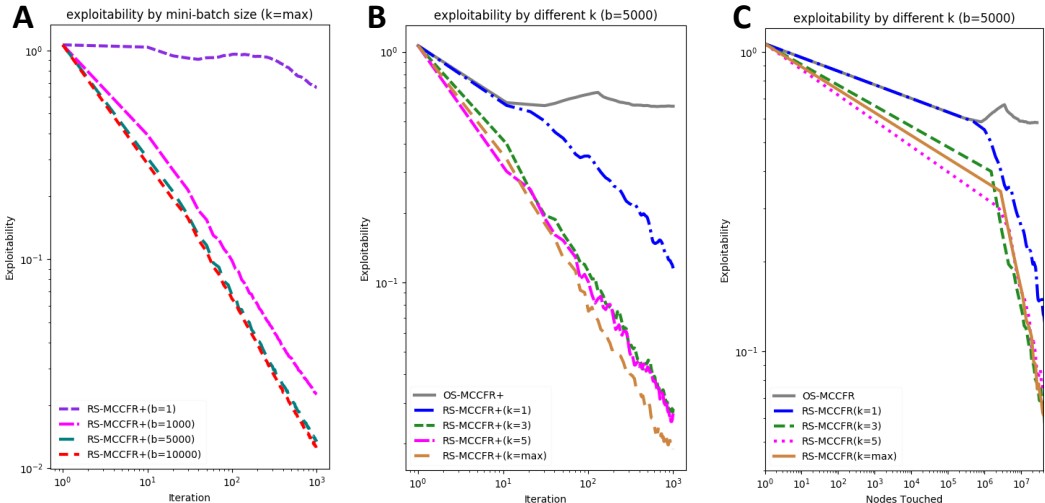

Figure 3: Comparison of different CFR-family methods in Leduc Hold'em. (A) Performance of robust sampling with different batch size. (B) Performance of robust sampling with different parameter $k$ by iteration. (C) Performance by the number of touched node.

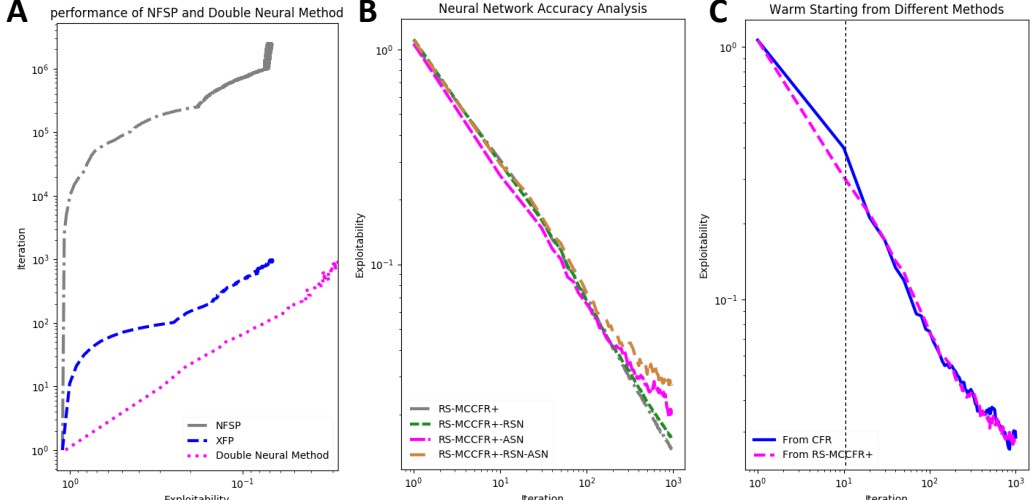

Figure 4: Performance of different methods in Leduc Hold'em. (A) comparison of NSFP, XFP and the proposed double neural method. (B) each contribution of RSN and ASN. (C) continue improvement from tabular based CFR and RS-MCCFR+.

**How does double neural CRF compare to the tabular counterpart, XFP, and NFSP?** To obtain an approximation of Nash equilibrium, Figure 4(A) demonstrates that NFSP needs $10^6$ iterations to reach a 0.06-Nash equilibrium, and requires $2 \times 10^5$ state-action pair samples and $2 \times 10^6$ samples for supervised learning respectively. The XFP needs $10^3$ iterations to obtain the same exploitability, however, this method is the precursor of NFSP and updated by a tabular based full-width fictitious play. Our proposed neural method only needs 200 iterations to achieve the same performance which shows that the proposed double neural algorithm converges significantly better than the reinforcement learning counterpart. In practice, our double neural method can achieve an exploitability of 0.02 after 1000 iterations, which is similar to the tabular method.

**What is the individual effect of RSN and ASN?** Figure 4(B) presents ablation study of the effects of RSN and ASN network respectively. Both MCCFR+-RSN and MCCFR+-ASN, which only employ one neural network, perform only slightly better than the double neural method. All the proposed neural methods can match the performance of the tabular based method.

**How well does continual improvement work?** In practice, we usually want to continually improve our strategy profile from an existing checkpoint (Brown & Sandholm, 2016). In the framework of the proposed neural counterfactual regret minimization algorithm, warm starting is easy and friendly. Firstly, we employ two neural networks to clone the existing tabular based cumulative

regret and the numerator of average strategy by optimizing Eq. (10). Then the double neural methods can continually improve the tabular based methods. As shown in Figure 4(C), warm start from either full-width based or sampling based CFR the existing can lead to continual improvements. Specifically, the first 10 iterations are learned by tabular based CFR and RS-MCCFR+. The remaining iterations are continually improved by the double neural method, where $b = 5000, k = max$.

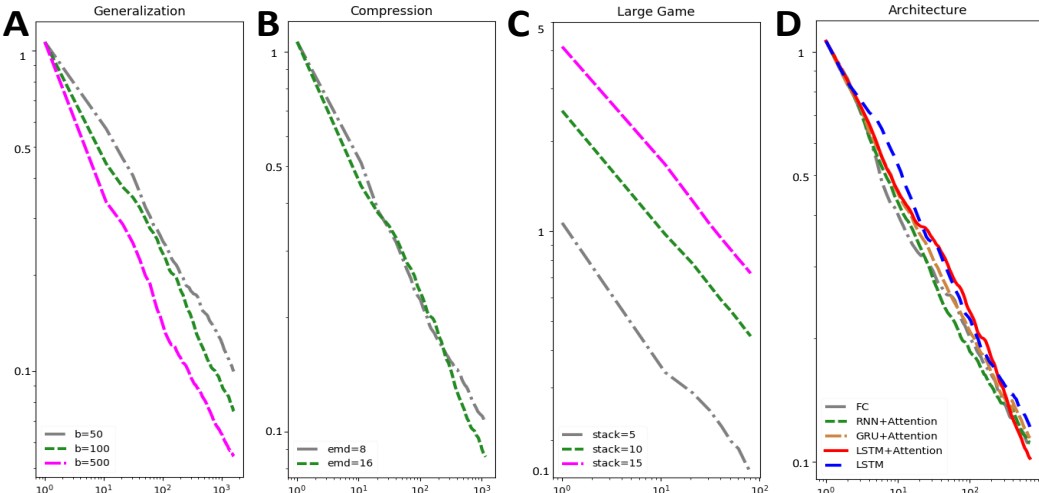

Figure 5: Performance analysis from different perspectives: (A) Generalization: by observed nodes. (B) Compression: by embedding size. (C) Large Game: by game size. (D) Architecture: by attention or not.

**Do the neural networks generalize to unseen information sets?** To investigate the generalization ability, we perform the neural CFR with small mini-batch sizes (b=50, 100, 500), where only $3.08\%, 5.59\%$, and $13.06\%$ information sets are observed in each iteration. In all these settings, the double neural can still converge and arrive at exploitability less than 0.1 within only 1000 iterations (Figure 5(A)).

**Do the neural networks just memorize but not generalize?** One indication that the neural networks are generalizing is that they use much fewer parameters than their tabular counterparts. We experimented with LSTM plus attention networks, and embedding size of 8 and 16 respectively. These architectures contain 1048 and 2608 parameters respectively in NLLH(5), both of which are much less than the tabular memory (more than $10^4$ number here). Note that both these two embedding sizes still leads to a converging strategy profile as shown in Figure 5(B).

**Does neural method converge in the larger game?** Figure 5(C) presents the log-log convergence curve of NLLH with different stack size (5, 10 and 15 respectively). The largest game size contains over $2 \times 10^7$ states and $3.7 \times 10^6$ information sets. Let mini-batch size be 500, there are $13.06\%$, $2.39\%$ and $0.53\%$ information sets that are observed respectively in each iteration. Even though only a small subset of nodes are sampled, the double neural method can still converge.

**Is attention in the neural architecture helpful?** Figure 5(D) presents the convergence curves of several different deep neural architectures, such as a fully connected deep neural network(FC), LSTM, LSTM plus attention, original RNN plus attention, and GRU plus attention. The recurrent neural network plus attention helps us obtain better strategies rate than other architectures after hundreds of iterations.

## 6 CONCLUSION AND FUTURE WORK

In this paper, we present a novel double neural counterfactual regret minimization method to solve large imperfect information game, which has a strong generalization and compression ability and can match the performance of tabular based CFR approach. We also developed a new sampling technique which has lower variance than the outcome sampling, while being more memory efficient than the external sampling. In the future, we plan to explore much more flexible methods and apply the double neural method to larger games, such as No-Limit Texas Hold'em.

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

## APPENDIX A   GAME RULES

**Leduc Hold'em** a two-players IIG of poker, which was first introduced in (Southey et al., 2012). In Leduc Hold'em, there is a deck of 6 cards comprising two suits of three ranks. The cards are often denoted by king, queen, and jack. In No-Limit Leduc Hold'em(**NLLH**), the player may wager any amount of chips up to a maximum of that player's remaining stack. There is also no limit on the number of raises or bets in each betting round. There are two rounds. In the first betting round, each player is dealt one card from a deck of 6 cards. In the second betting round, a community (or public) card is revealed from a deck of the remaining 4 cards. In this paper, we use NLLH(x) refer to the No-Limit Leduc Hold'em, whose stack size is $x$.

**One-Card Poker** is a two-players IIG of poker described by (Gordon, 2005). The game rules are defined as follows. Each player is dealt one card from a deck of $X$ cards. The first player can pass or bet, If the first player bet, the second player can call or fold. If the first player pass, the second player can pass or bet. If second player bet, the first player can fold or call. The game ends with two pass, call, fold. The fold player will lose 1 chips. If the game ended with two passes, the player with higher card win 1 chips, If the game end with call, the player with higher card win 2 chips.

## APPENDIX B   DEFINITION OF EXTENSIVE-FORM GAMES

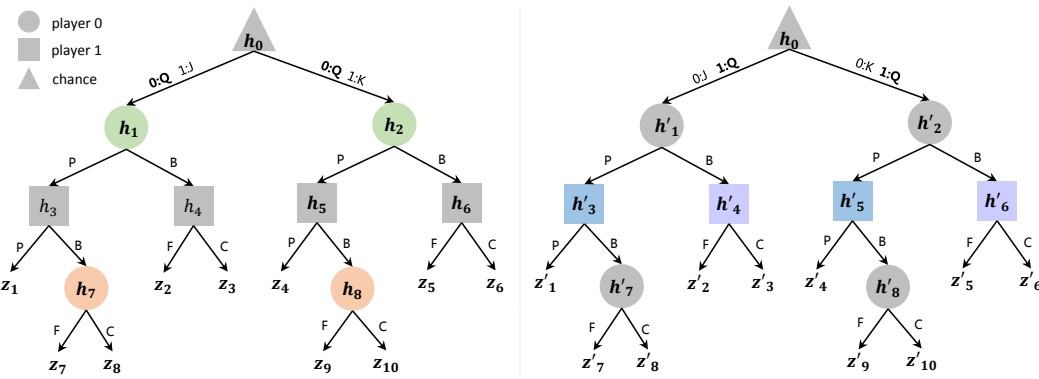

Figure 6: Illustration of extensive-form game. The left and right denote two different kinds of dealt private cards. We use same color other than gray for each state in the same information set. F, C, P, B refer to fold, call, pass, bet respectively.

### B.1   ADDITIONAL DEFINITIONS

For player $i$, the **expected game utility** $u_i^\sigma = \sum_{z \in Z} \pi^\sigma(z) u_i(z)$ of $\sigma$ is the expected payoff of all possible terminal nodes. Given a fixed strategy profile $\sigma_{-i}$, any strategy $\sigma_i^* = \max_{\sigma_i' \in \Sigma_i} u_i^{(\sigma_i', \sigma_{-i})}$ of player $i$ that achieves maximize payoff against $\pi_{-i}^\sigma$ is a **best response**. For two players' extensive-form games, a **Nash equilibrium** is a strategy profile $\sigma^* = (\sigma_0^*, \sigma_1^*)$ such that each player's strategy is a best response to the opponent. An $\epsilon$**-Nash equilibrium** is an approximation of a Nash equilibrium, whose strategy profile $\sigma^*$ satisfies: $\forall i \in N, u_i^{\sigma_i^*} + \epsilon \geq \max_{\sigma_i' \in \Sigma_i} u_i^{(\sigma_i', \sigma_{-i})}$.
**Exploitability** of a strategy $\sigma_i$ is defined as $\epsilon_i(\sigma_i) = u_i^{\sigma^*} - u_i^{(\sigma_i, \sigma_{-i}^*)}$. A strategy is unexploitable if $\epsilon_i(\sigma_i) = 0$. In large two player zero-sum games such poker, $u_i^{\sigma^*}$ is intractable to compute. However, if the players alternate their positions, the value of a pair of games is zeros, $i.e.$, $u_0^{\sigma^*} + u_1^{\sigma^*} = 0$ .
We define the exploitability of strategy profile $\sigma$ as $\epsilon(\sigma) = (u_1^{(\sigma_0, \sigma_1^*)} + u_0^{(\sigma_0^*, \sigma_1)})/2$.

### B.2   EXPLANATION BY EXAMPLE

To provide a more detailed explanation, Figure 6 presents an illustration of a partial game tree in One-Card Poker. In the first tree, two players are dealt (queen, jack) as shown in the left subtree and (queen, king) as shown in the right subtree. $z_i$ denotes terminal node and $h_i$ denotes non-terminal node. There are 19 distinct nodes, corresponding 9 non-terminal nodes including chance $h_0$ and 10 terminal nodes in the left tree. The trajectory from the root to each node is a history of actions. In an extensive-form game, $h_i$ refers to this history. For example, $h_3$ consists of actions 0:Q, 1:J and P.

$h_7$ consists of actions 0:Q, 1:J, P and B. $h_8$ consists of actions 0:Q, 1:K, P and B. We have $h_3 \sqsubseteq h_7$, $A(h_7) = \{P, B\}$ and $P(h_3) = 1$

In IIG, the private card of player 1 is invisible to player 0, therefore $h_7$ and $h_8$ are actually the same for player 0. We use information set to denote the set of these undistinguished states. Similarly, $h_1$ and $h_2$ are in the same information set. For the right tree of Figure 6, $h_3'$ and $h_5'$ are in the same information set. $h_4'$ and $h_6'$ are in the same information set.

Generally, any $I_i \in \mathcal{I}$ could only remember the information observed by player $i$ including player $i's$ hidden variable and public actions. For example, the information set of $h_7$ and $h_8$ indicates a sequence of 0:Q, P, and B. Because $h_7$ and $h_8$ are undistinguished by player 0 in IIG, all the states have a same strategy. For example, $I_0$ is the information set of $h_7$ and $h_8$, we have $I_0 = I_0(h_7) = I_0(h_8)$, $\sigma_0(I_0) = \sigma_0(h_7) = \sigma_0(h_8)$, $\sigma_0(a|I_0) = \sigma_0(a|h_7) = \sigma_0(a|h_8)$.

## APPENDIX C    ADDITIONAL EXPERIMENT DETAILS

### C.1    FEATURE ENCODING OF POKER GAMES

**The feature is encoded as following.** As shown in the figure 2 (A), for a history $h$ and player $P(h)$, we use one-hot encoding (Harris & Harris) to represent the observed actions including chance player. For example, the input feature $x_l$ for $l$-th cell is the concatenation of three one-hot features including the given private cards, the revealed public cards and current action $a$. Both the private cards and public cards are encoded by one-hot technique, where the value in the existing position is 1 and the others are 0. If there are no public cards, the respective position will be filled with 0. Because the action taking by chance is also a cell in the proposed sequential model. Thus in a No-Limit poker, such as Leduc Hold'em, action $a$ could be any element in $\{$fold, cumulative spent$\} \cup$ $\{$public cards$\}$ , where cumulative spent denotes the total chips after making a call or raise. The length of the encoding vector of action $a$ is the quantities of public cards plus 2, where cumulative spent is normalized by the stack size.

### C.2    ADDITIONAL EXPERIMENT RESULTS

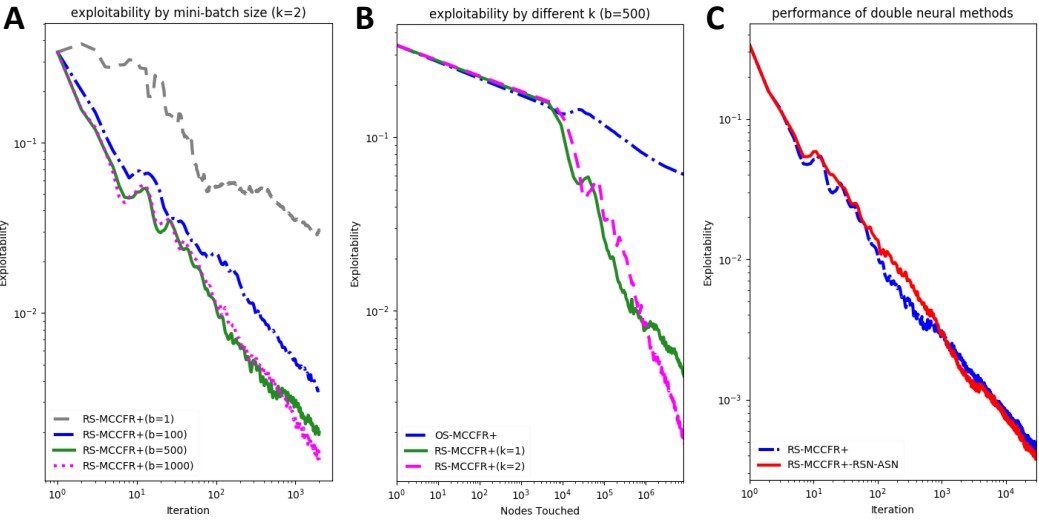

Figure 7: Comparison of different CFR-family methods and neural network methods in One-Card-Poker. (A) Comparison of the robust sampling with different mini-batch size. (B) Comparison of the outcome sampling and the robust sampling with different sample actions k. (C) Comparison of tabular based RS-MCCFR+ and the double neural method.

Figure 7 (A) presents the convergence rate for the proposed robust sampling method of different mini-batch size $b = (1, 100, 500, 1000)$. The experimental results are similar to Leduc Hold'em poker, larger mini-batch size indicates a better exploitability.

Figure 7 (B) demonstrates that the convergence rate for different sampling methods including outcome sampling and robust sampling under $k = 1, 2$. The conclusion is that RS-MCCFR+

converges significantly faster than OS-MCCFR+ after touching the same number of nodes. Experiment results show that $k = 1$ has a similar trend with $k = 2$ (external sampling). Because only one trajectory is sampled, the proposed RS-MCCFR+ will require less memory than the external sampling.

Figure 7 (C) compares the performance between the tabular method and the double neural method. Experimental results demonstrate that RS-MCCFR+-RSN-ASN can achieve an exploitability of less than 0.0004 in One-Card Poker, which matches the performance of the tabular method. For RSN and ASN, we set neural batch size 4, hidden size 32 and learning rate 0.001.

## APPENDIX D    DETAILS OF RECURRENT NEURAL NETWORK

In order to define our $\mathcal{R}$ and $\mathcal{S}$ network, we need to represent the information set $I_i \in \mathcal{I}$ in extensive-form games. In such games, players take action in alternating fashion and each player makes a decision according to the observed history. In this paper, we model the behavior sequence as a recurrent neural network and each action in the sequence corresponds to a cell in RNN. Figure 2 (A) provides an illustration of the proposed deep sequential neural network representation for information sets.

In standard RNN, the recurrent cell will have a very simple structure, such as a single tanh or sigmoid layer. Hochreiter & Schmidhuber (1997) proposed a long short-term memory method (LSTM) with the gating mechanism, which outperforms the standard version and is capable of learning long-term dependencies. Thus we will use LSTM for the representation. Furthermore, different position in the sequence may contribute differently to the decision making, we will add an attention mechanism (Desimone & Duncan, 1995; Cho et al., 2015) to the LSTM architecture to enhance the representation. For example, the player may need to take a more aggressive strategy after beneficial public cards are revealed. Thus the information, after the public cards are revealed may be more important.

More specifically, for $l$-th cell, define $x_l$ as the input vector (which can be either player or chance actions), $e_l$ as the hidden layer embedding, $\phi_*$ as a general nonlinear function. Each action is represented by a LSTM cell, which has the ability to remove or add information to the cell state with three different gates. Define the notation $\cdot$ as element-wise product. The first **forgetting gate** layer is defined as $g_l^f = \phi_f(w^f[x_l, e_{l-1}])$, where $[x_l, e_{l-1}]$ denotes the concatenation of $x_l$ and $e_{l-1}$. The second **input gate** layer decides which values to update and is defined as $g_l^i = \phi_i(w^i[x_l, e_{l-1}])$. A nonlinear layer output a vector of new candidate values $\tilde{C}_l = \phi_c(w^l[x_l, e_{l-1}])$ to decide what can be added to the state. After the forgetting gate and the input gate, the new cell state is updated by $C_l = g_l^f \cdot C_{l-1} + g_l^i \cdot \tilde{C}_l$. The third **output gate** is defined as $g_l^o = \phi_o(w^o[x_l, e_{l-1}])$. Finally, the updated hidden embedding is $e_l = g_l^o \cdot \phi_e(C_l)$. As shown in Figure 2 (A), for each LSTM cell $j$, the vector of attention weight is learned by an **attention network**. Each member in this vector is a scalar $\alpha_j = \phi_a(w^a e_j)$. The attention embedding of $l$-th cell is then defined as $e_l^a = \sum_{j=1}^l \alpha_j \cdot e_j$, which is the summation of the hidden embedding $e_j$ and the learned attention weight $\alpha_j$. The final output of the network is predicted by a **value network**, which is defined as

$$\tilde{y}_l := f(a, I_i|\theta) = w^y \phi_v(e_l^a) = w^y \phi_v \left( \sum_{j=1}^l \phi_a(w^a e_j) \cdot e_j \right), \tag{15}$$

where $\theta$ is the parameters in the defined sequential neural networks. Specifically, $\phi_f$, $\phi_i$, $\phi_o$ are sigmoid functions. $\phi_c$ and $\phi_e$ are hyperbolic tangent functions. $\phi_a$ and $\phi_v$ are rectified linear functions. The proposed **RSN** and **ASN** share the same neural architecture, but use different parameters. That is $\mathcal{R}(a, I_i|\theta_{\mathcal{R}}^t) = f(a, I_i|\theta_{\mathcal{R}}^t)$ and $\mathcal{S}(a, I_i|\theta_{\mathcal{S}}^t) = f(a, I_i|\theta_{\mathcal{S}}^t)$. $\mathcal{R}(\cdot, I_i|\theta_{\mathcal{R}}^t)$ and $\mathcal{S}(\cdot, I_i|\theta_{\mathcal{S}}^t)$ denote two vectors of inference value for all $a \in A(I_i)$.

## APPENDIX E    NEURAL AGENT FOR OPTIMIZING NEURAL REPRESENTATION

Define $\beta_{epoch}$ as training epoch, $\beta_{lr}$ as learning rate, $\beta_{loss}$ as the criteria for early stopping, $\beta_{re}$ as the upper bound for the number of iterations from getting the minimal loss last time, $\theta^{t-1}$ as the parameter to optimize, $f(\cdot|\theta^{t-1})$ as the neural network, $\mathcal{M}$ as the training sample consisting information set and the corresponding target. To simplify notations, we use $\beta^*$ to denote the set of hyperparameters in the proposed deep neural networks. $\beta_{\mathcal{R}}^*$ and $\beta_{\mathcal{S}}^*$ refer to the

---

**Algorithm 2:** Optimization of Deep Neural Network

1  **Function** NeuralAgent ($f(\cdot|\theta^{T-1})$, $\mathcal{M}$, $\theta^{T-1}$, $\beta^*$):
2     initialize $optimizer$, $scheduler$         ▷ gradient descent optimizer and learning rate scheduler
3     $\theta^T \leftarrow \theta^{T-1}$, $l_{best} \leftarrow \infty$, $t_{best} \leftarrow 0$       ▷ warm starting from the checkpoint of the last iteration
4     **For** $t = 1$ *to* $\beta_{epoch}$ **do**
5         $loss \leftarrow []$               ▷ initialize $loss$ as an empty list
6         **For** *each training epoch* **do**
7             $\{x^{(i)}, y^{(i)}\}_{i=1}^m \sim \mathcal{M}$       ▷ sampling a mini-batch from $\mathcal{M}$
8             $batch\_loss \leftarrow \frac{1}{m} \sum_{i=1}^m (f(x^{(i)}|\theta^{T-1}) + y^{(i)} - f(x^{(i)}|\theta^T))^2$
9             back propagation $batch\_loss$ with learning rate $lr$
10            clip gradient of $\theta^T$ to $[-\epsilon, \epsilon]^d$       ▷ $d$ is the dimension of $\theta^T$
11            $optimizer(batch\_loss)$
12            $loss.append(batch\_loss)$
13         $lr \leftarrow sheduler(lr)$       ▷ reduce learning rate adaptively when loss has stopped improving
14         **if** $avg(loss) < \beta_{loss}$ **then**
15            $\theta_{best}^T \leftarrow \theta^T$, early stopping.       ▷ if loss is small enough, using early stopping mechanism.
16         **else if** $avg(loss) < l_{best}$ **then**
17            $l_{best} = avg(loss)$, $t_{best} \leftarrow t$, $\theta_{best}^T \leftarrow \theta^T$
18         **if** $t - t_{best} > \beta_{re}$ **then**
19            $lr \leftarrow \beta_{lr}$       ▷ reset learning rate to escape from potential saddle point or local minima.
20     **return** $\theta^T$

---

sets of hyperparameters in RSN and ASN respectively. According to our experiments, we find a carefully designed optimization method can help us obtain a relatively higher convergence rate of exploitability. Algorithm 2 presents the details of how to optimize the proposed neural networks.

Both $\mathcal{R}(a, I_i|\theta_{\mathcal{R}}^{t+1})$ and $\mathcal{S}(a, I_i|\theta_{\mathcal{S}}^t)$ are optimized by mini-batch stochastic gradient descent method. In this paper, we use Adam optimizer (Kingma & Ba, 2014) with both momentum and adaptive learning rate. Some other optimizers such as Nadam, RMSprop, Nadam from (Ruder, 2017) are also tried in our experiments, however, they do not achieve better experimental results. In practice, existing optimizers may not return a relatively low enough loss because of potential saddle point or local minima. To obtain a relatively higher accuracy and lower optimization loss, we use a carefully designed scheduler to reduce the learning rate when the loss has stopped decrease. Specifically, the scheduler reads a metrics quantity, $e.g$, mean squared error, and if no improvement is seen for a number of epochs, the learning rate is reduced by a factor. In addition, we will reset the learning rate in both optimizer and scheduler once loss stops decrease in $\beta_{re}$ epochs. Gradient clipping mechanism is used to limit the magnitude of the parameter gradient and make optimizer behave better in the vicinity of steep cliffs. After each epoch, the best parameter will be updated. Early stopping mechanism is used once the lowest loss is less than the specified criteria $\beta_{loss}$.

In experiments, we set the network hyperparameters as follow. For RSN, we set the hyperparameters as follows: neural batch size is 256 and learning rate $\beta_{lr} = 0.001$. A scheduler, who will reduce the learning rate based on the number of epochs and the convergence rate of loss, help the neural agent to obtain a high accuracy. The learning rate will be reduced by 0.5 when loss has stopped improving after 10 epochs. The lower bound on the learning rate of all parameters in this scheduler is $10^{-6}$. To avoid the algorithm converging to potential local minima or saddle point, we will reset the learning rate to 0.001 and help the optimizer to learn a better performance. $\theta_{best}^T$ is the best parameters to achieve the lowest loss after $T$ epochs. If average loss for epoch $t$ is less than the specified criteria $\beta_{loss}=10^{-4}$, we will early stop the optimizer. We set $\beta_{epoch} = 2000$ and update the optimizer 2000 maximum epochs. For ASN, we set the loss of early stopping criteria as $10^{-5}$. The learning rate will be reduced by 0.7 when loss has stopped improving after 15 epochs. Other hyperparameters in ASN are similar to RSN.

## APPENDIX F  OPTIMIZE COUNTERFACTUAL REGRET MINIMIZATION WITH TWO DEEP NEURAL NETWORKS

---

**Algorithm 3:** Mini-Batch RS-MCCFR with Double Neural Networks

---

1 **Function** `Mini-Batch-MCCFR-NN`($t$):
2 $\quad$ $\mathcal{M}_{\mathcal{R}} \leftarrow \emptyset, \mathcal{M}_{\mathcal{S}} \leftarrow \emptyset$
3 $\quad$ **For all** $i = 1$ *to* $b$ **do in parallel then**
4 $\quad\quad$ MCCFR-NN($t, \emptyset, 0, 1, 1$)
5 $\quad\quad$ MCCFR-NN($t, \emptyset, 1, 1, 1$)
6 $\quad$ **return** $\mathcal{M}_{\mathcal{R}}, \mathcal{M}_{\mathcal{S}}$

7

8 **Function** `MCCFR-NN`($t, h, i, \pi_i, \pi_i^{rs(k)}$):
9 $\quad$ $I_i \leftarrow I_i(h)$ $\hfill \triangleright$ information set at state $h$
10 $\quad$ **if** $h \in Z$ **then**
11 $\quad\quad$ **return** $\frac{u_i(h)}{\pi_i^{rs(k)}}$ $\hfill \triangleright$ return game payoff
12 $\quad$ **else if** $P(h) = -1$ **then**
13 $\quad\quad$ $a \sim \sigma_{-i}(I_i)$ $\hfill \triangleright$ Sample an action from $\sigma_{-i}(h)$
14 $\quad\quad$ **return** MCCFR-NN($t, ha, i, \pi_i, \pi_i^{rs(k)}$)
15 $\quad$ **else if** $P(h) = i$ **then**
16 $\quad\quad$ $\hat{R}_i(\cdot|I_i) \leftarrow \mathcal{R}(\cdot, I_i|\theta_{\mathcal{R}}^t)$ **if** $t > 1$ **else** $\vec{0}$ $\hfill \triangleright$ inference the vector of cumulative regret $\forall a \in A(I_i)$
17 $\quad\quad$ $\sigma_i(I_i) \leftarrow$ CalculateStrategy($\hat{R}_i(\cdot|I_i), I_i$) $\hfill \triangleright$ calculate current strategy
18 $\quad\quad$ $v_i(h) \leftarrow 0, r_i(\cdot|I_i) \leftarrow \vec{0}, s_i(\cdot|I_i) \leftarrow \vec{0}$ $\hfill \triangleright$ $r_i(\cdot|I_i)$ and $_i(\cdot|I_i)$ are two vectors over $A(I_i)$
19 $\quad\quad$ $A^{rs(k)}(I_i) \leftarrow$ sampling $k$ different actions according to $\sigma_i^{rs(k)}$
20 $\quad\quad$ **For** $a \in A^{rs(k)}(I_i)$ **do**
21 $\quad\quad\quad$ $v_i(a|h) \leftarrow$ MCCFR-NN($t, ha, i, \pi_i\sigma_i(a|I_i), \pi_i^{rs}\sigma_i^{rs(k)}(a|I_i)$)
22 $\quad\quad\quad$ $v_i(h) \leftarrow v_i(h) + v_i(a|h)\sigma_i(a|I_i)$ $\hfill \triangleright$ update counterfactual value
23 $\quad\quad$ **For** $a \in A^{rs(k)}(I_i)$ **do**
24 $\quad\quad\quad$ $r_i(a|I_i) \leftarrow v_i(a|h) - v_i(h)$ $\hfill \triangleright$ update cumulative regret
25 $\quad\quad\quad$ $s_i(a|I_i) \leftarrow \pi_i^\sigma(I_i)\sigma_i(a|I_i)$ $\hfill \triangleright$ update average strategy numerator
26 $\quad\quad$ Store updated cumulative regret tuple $(I_i, r_i(\cdot|I_i))$ in $\mathcal{M}_{\mathcal{R}}$
27 $\quad\quad$ Store updated current strategy dictionary $(I_i, s_i(\cdot|I_i))$ in $\mathcal{M}_{\mathcal{S}}$
28 $\quad$ **else**
29 $\quad\quad$ $\hat{R}_{-i}(\cdot|I_i) \leftarrow \mathcal{R}(\cdot, I_i|\theta_{\mathcal{R}}^t)$ **if** $t > 1$ **else** $\vec{0}$ $\hfill \triangleright$ inference cumulative regret
30 $\quad\quad$ $\sigma_{-i}(I_i) \leftarrow$ CalculateStrategy($\hat{R}_{-i}(\cdot|I_i), I_i$) $\hfill \triangleright$ calculate current strategy
31 $\quad\quad$ $a \sim \sigma_{-i}(I_i)$ $\hfill \triangleright$ Sample an action from $\sigma_{-i}(I_i)$
32 $\quad\quad$ **return** MCCFR-NN($t, ha, i, \pi_i, \pi_i^{rs(k)}$)

33

34 **Function** `CalculateStrategy`($R_i(\cdot|I_i), I_i$):
35 $\quad$ $sum \leftarrow \sum_{a \in A(I_i)} \max(R_i(a|I_i), 0)$
36 $\quad$ **For** $a \in A(I_i)$ **do**
37 $\quad\quad$ $\sigma_i(a|I_i) = \frac{\max(R_i(a|I_i), 0)}{sum}$ **if** sum $> 0$ **else** $\frac{1}{|A(I_i)|}$
38 $\quad$ **return** $\sigma_i(I_i)$

39

---

Algorithm 3 presents one application scenario of the proposed double neural method, which is based on the proposed mini-batch robust sampling method. The function MCCFR-NN will traverse the game tree like tabular MCCFR, which starts from the root history $h = \emptyset$. Define $I_i$ as the information set of $h$. Suppose that player $i$ will sample $k$ actions according to the robust sampling. Then the function can be defined as follows. (1) If the history is terminal, the function returns the weighted utility. (2) If the history is the chance player, one action $a \in A(I_i)$ will be sampled according to the strategy $\sigma_{-i}(I_i)$. Then this action will be added to the history, *i.e.*, $h \leftarrow ha$. (3) If $P(I_i) = i$,

the current strategy can be updated by the cumulative regret predicted by RSN. Then we sample $k$ actions according the specified sampling strategy profile $\sigma_i^{rs(k)}$. After a recursive updating, we can obtain the counterfactual value and regret of each action at $I_i$. For the visited node, their counterfactual regrets and numerators of the corresponding average strategy will be stored in $\mathcal{M}_{\mathcal{R}}$ and $\mathcal{M}_{\mathcal{S}}$ respectively. (4) If $P(I_i)$ is the opponent, only one action will be sampled according the strategy $\sigma_{-i}(I_i)$.

The function Mini-Batch-MCCFR-NN presents a mini-batch sampling method, where $b$ blocks will be sampled in parallel. This mini-batch method can help the MCCFR to achieve a more accurate estimation of CFV. The parallel sampling makes this method efficient in practice.

## APPENDIX G    THEORETICAL ANALYSIS

### G.1    REACH PROBABILITY AND POSTERIOR PROBABILITY

**Lemma 1:** The state reach probability of one player is proportional to posterior probability of the opponent's hidden variable, $i.e.$, $p(h_{-i}^v|I_i) \propto \pi_{-i}^\sigma(h)$.

**Proof:** For player $i$ at information set $I_i$ and fixed $i's$ strategy profile $\sigma_i$, $i.e.$, $\forall h \in I_i, \pi_i^\sigma(h)$ is constant. Based on the defination of extensive-form game in Section 2.1, the cominbation of $I_i$ and opponent's hidden state $h_{-i}^v$ can indicate a particular history $h = h_i^v h_{-i}^v a_0 a_1 ... a_{L-1}$. With Bayes' Theorem, we can inference the posterior probability of opponent's private cards with Equation16

$$
\begin{aligned}
p(h_{-i}^v|I_i) &= \frac{p(h_{-i}^v, I_i)}{p(I_i)} = \frac{p(h)}{p(I_i)} \propto p(h) \\
&\propto p(h_i^v)p(h_{-i}^v) \prod_{l=1}^{L} \sigma_{P(h_i^v h_{-i}^v a_0 a_1 ... a_{l-1})}(a_l|h_i^v h_{-i}^v a_0 a_1 ... a_{l-1}) \\
&\propto \pi^\sigma(h) = \pi_i^\sigma(h)\pi_{-i}^\sigma(h) \\
&\propto \pi_{-i}^\sigma(h)
\end{aligned}
\tag{16}
$$

### G.2    ROBUST SAMPLING, OUTCOME SAMPLING AND EXTERNAL SAMPLING

**Lemma 3:** If $k = max_{I_i \in \mathcal{I}}|A(I_i)|$ and $\forall i \in N, \forall I_i \in \mathcal{I}_i, \forall a \in A(I_i), \sigma_i^{rs(k)}(a|I_i) \sim unif(0, |A(I_i)|)$, then robust sampling is same with external sampling. (see the proof in section G.2)

**Lemma 4:** If $k = 1$ and $\sigma_i^{rs(k)} = \sigma_i$, then robust sampling is same with outcome sampling. (see the proof in section G.2)

For robust sampling, given strategy profile $\sigma$ and the sampled block $Q_j$ according to sampling profile $\sigma^{rs(k)} = (\sigma_i^{rs(k)}, \sigma_{-i})$, then $q(z) = \pi_i^{\sigma^{rs(k)}}(z)\pi_{-i}^\sigma(z)$, and the regret of action $a \in A^{rs(k)}(I_i)$ is

$$
\begin{aligned}
\tilde{r}_i^\sigma((a|I_i)|Q_j) &= \tilde{v}_i^\sigma((a|I_i)|Q_j) - \tilde{v}_i^\sigma(I_i|Q_j) \\
&= \sum_{z \in Q_j, ha \sqsubseteq z, h \in I_i} \frac{1}{q(z)}\pi_{-i}^\sigma(z)\pi_i^\sigma(ha, z)u_i(z) - \sum_{z \in Q_j, h \sqsubseteq z} \frac{1}{q(z)}\pi_{-i}^\sigma(z)\pi_i^\sigma(h, z)u_i(z) \\
&= \sum_{z \in Q_j, ha \sqsubseteq z, h \in I_i} \frac{u_i(z)}{\pi_i^{\sigma^{rs(k)}}(z)}\pi_i^\sigma(ha, z) - \sum_{z \in Q_j, h \sqsubseteq z, h \in I_i} \frac{u_i(z)}{\pi_i^{\sigma^{rs(k)}}(z)}\pi_i^\sigma(h, z) \\
&= \sum_{z \in Q_j, ha \sqsubseteq z, h \in I_i} \pi_i^\sigma(ha, z)u_i^{rs}(z) - \sum_{z \in Q_j, h \sqsubseteq z, h \in I_i} \pi_i^\sigma(h, z)u_i^{rs}(z),
\end{aligned}
\tag{17}
$$

where $u_i^{rs}(z) = \frac{u_i(z)}{\pi_i^{\sigma^{rs(k)}}(z)}$ is the weighted utility according to reach probability $\pi_i^{\sigma^{rs(k)}}(z)$. Because the weighted utility no long requires explicit knowledge of the opponent's strategy, we can use this sampling method for online regret minimization.

Generally, if player $i$ randomly selects $min(k, |A(I_i)|)$ actions according to discrete uniform distribution $unif(0, |A(I_i)|)$ at information set $I_i$, i.e., $\sigma_i^{rs(k)}(a|I_i) = \frac{min(k,|A(I_i)|)}{|A(I_i)|}$, then

$$\pi_i^{\sigma^{rs(k)}}(I_i) = \prod_{h \in I_i, h' \sqsubseteq h, h'a \sqsubseteq h, h' \in I_i'} \frac{min(k, |A(I_i')|)}{|A(I_i')|} \tag{18}$$

and $u_i^{rs}(z)$ is a constant number when given the sampling profile $\sigma^{rs(k)}$.
Specifically,

- if $k = max_{I_i \in I}|A(I_i)|$, then $\sigma_i^{rs(k)}(I_i) = 1$, $u_i^{rs(k)}(z) = u_i(z)$, and

$$\tilde{r}_i^\sigma((a|I_i)|Q_j) = \sum_{z \in Q_j, h \sqsubseteq z, h \in I_i} u_i(z)(\pi_i^\sigma(ha, z) - \pi_i^\sigma(h, z)) \tag{19}$$

  Therefore, **robust sampling is same with external sampling when $k = max_{I_i \in I}|A(I_i)|$.**

- if $k = 1$ and $\sigma_i^{rs(k)} = \sigma_i$, only one history $z$ is sampled in this case,then $u_i^{rs(k)}(z) = \frac{u_i(z)}{\pi_i^{\sigma_i}(z)}$, $\exists h \in I_i$, for $a \in A^{rs(k)}(I_i)$

$$\begin{aligned} \tilde{r}_i^\sigma((a|I_i)|Q_j) &= \tilde{r}_i^\sigma((a|h)|Q_j) \\ &= \sum_{z \in Q_j, ha \sqsubseteq z} \pi_i^\sigma(ha, z)u_i^{rs}(z) - \sum_{z \in Q_j, h \sqsubseteq z} \pi_i^\sigma(h, z)u_i^{rs}(z) \\ &= \frac{(1 - \sigma_i(a|h))u_i(z)}{\pi_i^\sigma(ha)} \end{aligned} \tag{20}$$

  For $a \notin A^{rs(k)}(I_i)$, the regret will be $\tilde{r}_i^\sigma((a|h)|j) = 0 - \tilde{v}_i^\sigma(h|j)$. Therefore, **robust sampling is same with outcome sampling when $k = 1$ and $\sigma_i^{rs(k)} = \sigma_i$.**

- if $k = 1$, and player $i$ randomly selects one action according to discrete uniform distribution $unif(0, |A(I_i)|)$ at information set $I_i$, then $u_i^{rs(1)}(z) = \frac{u_i(z)}{\pi_i^{\sigma^{rs(k)}}(z)}$ is a constant, $\exists h \in I_i$, for $a \in A^{rs(k)}(I_i)$

$$\begin{aligned} \tilde{r}_i^\sigma((a|I_i)|Q_j) &= \sum_{z \in Q_j, ha \sqsubseteq z, h \in I_i} \pi_i^\sigma(ha, z)u_i^{rs}(z) - \sum_{z \in Q_j, h \sqsubseteq z, h \in I_i} \pi_i^\sigma(h, z)u_i^{rs}(z) \\ &= (1 - \sigma_i(a|h))\pi_i^\sigma(ha, z)u_i^{rs(1)}(z) \end{aligned} \tag{21}$$

  if action $a$ is not sampled at state $h$, the regret is $\tilde{r}_i^\sigma((a|h)|j) = 0 - \tilde{v}_i^\sigma(h|j)$. **Compared to outcome sampling, the robust sampling in that case have a lower variance** because of the constant $u_i^{rs(1)}(z)$.

### G.3 MINI-BATCH MCCFR GIVES AN UNBIASED ESTIMATION OF COUNTERFACTUAL VALUE

**Lemma 5:** $E_{Q^j \sim \text{Robust Sampling}}[\tilde{v}_i^\sigma(I_i|b)] = v_i^\sigma(I_i)$.

**Proof:**

$$E_{Q^j \sim \text{Robust Sampling}}[\tilde{v}_i^\sigma(I_i|b)] = E_{b' \sim \text{unif(0,b)}}[\tilde{v}_i^\sigma(I_i|b')]$$

$$= E_{b' \sim \text{unif(0, b)}} \left( \sum_{j=1}^{b'} \sum_{h \in I_i, z \in Q^j, h \sqsubseteq z} \frac{\pi_{-i}^\sigma(z)\pi_i^\sigma(h, z)u_i(z)}{q(z)b'} \right)$$

$$= E_{b' \sim \text{unif(0, b)}} \left( \frac{1}{b'} \sum_{j=1}^{b'} \tilde{v}_i^\sigma(I_i|Q^j) \right)$$

$$= \frac{1}{b} \sum_{b'=1}^{b} \left( \frac{1}{b'} \sum_{j=1}^{b'} \tilde{v}_i^\sigma(I_i|Q^j) \right) \tag{22}$$

$$= \frac{1}{b} \sum_{b'=1}^{b} \left( \frac{1}{b'} \sum_{j=1}^{b'} E(\tilde{v}_i^\sigma(I_i|Q^j)) \right)$$

$$= \frac{1}{b} \sum_{b'=1}^{b} \left( \frac{1}{b'} \sum_{j=1}^{b'} v_i^\sigma(I_i) \right)$$

$$= v_i^\sigma(I_i)$$

