# OpenReview forum: "Double Neural Counterfactual Regret Minimization"
_ICLR.cc/2019/Conference_

### Official Review · AnonReviewer1 · 2018-11-02
**Review: Interesting application of function approximation to CFR but requires larger experiments**

**Rating:** 4
**Confidence:** 5

**Review:**

========= Summary =========

The authors propose "Double Neural CFR", which uses neural network function approximation in place of the tabular update in CFR. CFR is the leading method for finding equilibria in imperfect information games. However it is typically employed with a tabular policy, limiting its applicability large games. Typically, hand-crafted abstractions are employed for games that are too large for exact tabular solutions. Function approximation could remove the necessity for hand-crafted abstractions and allow CFR to scale to larger problems.

The DN-CFR algorithm roughly consists of:
- start with an arbitrary vmodel_0, smodel_0
for t = 0,1,2,3:
  - collect a "batch" of (infoset I, immediate counterfactual value v_I) samples by traversal against vmodel_t (as well as I, strategy samples)
  - train a new network vmodel_{t+1} on this "batch" with y=(v_I + vmodel_t(I)) and MSE loss
  - similarly for (I, strategy)
- return smodel_t

The authors also propose a novel MC samping strategy this is a mixture between outcome and external sampling.

DN-CFR is evaluated on two games: a variant of Leduc hold-em with stack size 5, and one-card poker with 5 cards. If I understand correctly, these games have <10,000 and <100 infosets, respectively. The authors show that DN-CFR achieves similar convergence rates to tabular CFR, and outperform NFSP variants.

========== Comments ========

The authors are exploring an important problem that is of great interest to the IIG community. Their application of NN function approximation is reasonable and mostly theoretically well-grounded (but see below), I think it's on the right track. However, the games that are used for evaluation are very small, in fact I believe they have fewer states than the number of parameters in their network (the number of network parameters is not provided but I assume >1000). As a result, the NN is not providing any compression or generalization, and I would expect that the network can memorize the training set data exactly, i.e. predict the exact mean counterfactual value for each infoset over the data. If that's true, then DN-CFR is essentially exactly replicating tabular CFR (the approximation serves no purpose).

As a result, in my opinion this work fails to address the important challenges for function approximation in CFR, namely:

- Can function approximation allow for *generalization across infosets* in order to reduce sample complexity of CFR (i.e. an unsupervised abstraction)? Are specific network architectures required for good generalization?
- The magnitude of counterfactual regrets in the support of the equilibrium decays to zero relative to dominated actions. Are NN models capable to estimating the regrets accurately enough to converge to a good strategy?
- Are optimization methods able to deal with the high variance in large IIGs?
- Since each successive approximation is being trained from the previous NN, does this cause errors to accumulate?
- How do approximation errors accumulate across CFR iterations?
- Is minimizing MSE loss sufficient to approximate the strategy well? (since the mapping from regrets -> strategy is non-linear)

I believe there is also a theoretical problem with this algorithm. In Eq. 8, they minimize the loss of CF value predictions *over the distribution of infosets in the last CFR step ("batch")*. However, this distribution may change between CFR iterations, for example if the strat_t folds 2-3 on the preflop then flop infosets with 2-3 hole cards will never be observed on iteration t+1. As a result, the NN loss will only be minimized over infosets observed in the last iteration - so the network will "forget" the regrets for all other infosets. I think this issue does not arise in these toy games because all infosets are observed at each iteration, but this is certainly not the case in real games.
There are a number of ways that this issue could be addressed (e.g. train on historical infosets rather than the current batch, etc.) These would need to be explored.

I would recommend that the authors evaluate on more complex games to answer the important questions stated above and resubmit. I think this would also make an excellent workshop submission in its current state as it contains many interesting ideas.

Detailed comments:

"... the original CFR only works for discrete stand and action spaces...": Are the authors implying that DN-CFR addresses this limitation?
"Moravk" -> "Moravcik"
"...these methods do not explicitly take into account the hidden information in a game..." Could you clarify? Is your point that these methods operate on the normal form rather than extensive form game?
"care algorithm design" -> "careful algorithm design"
The paragraph starting "In standard RNN..." should be reduced or moved to appendix. The exact NN architecture is not central to the ideas, and there are no experimental comparison with other architectures so we have no evidence that the architecture is relevant.

---

> ### Author Response · Authors · 2018-11-23
> **Reply to "requires larger experiments and the questions" (Q1-Q6)**
>
> Thanks for your effort in providing this detailed and useful review!
>
> We present our clarification in the following and the common questions are answered in the public comment “Paper revision 1”:
>
> Q1: Can function approximation allow for *generalization across infosets* in order to reduce sample complexity of CFR (i.e. an unsupervised abstraction)? Q2: The games that are used for evaluation are very small, in fact I believe they have fewer states than the number of parameters in their network?
>
> A: In each iteration, we sampled less than 1% of the information sets, but we compute exploitability in all information sets. This indicates that neural network can generalize to those unseen information sets. We also conducted further experiments in the revised paper where even fewer information sets are sampled and smaller neural networks are used. We showed that the algorithms still converge in terms of exploitability. These new experiments also support the generalization ability of our double neural model. Further details please see  Figure 5(A), 5(B) and 5(C) in the revised paper.
>
> Q3: Are specific network architectures required for good generalization?
>
> A:  According to our experiments，network architectures and optimization methods are very important. We find a carefully designed architecture will help us achieve a faster convergence rate. Further details please see  Figure 5(D) in the revised paper.
>
>
> Q4: The magnitude of counterfactual regrets in the support of the equilibrium decays to zero relative to dominated actions. Are NN models capable to estimating the regrets accurately enough to converge to a good strategy?
>
> A: Yes.  As the increasing of the number of iterations T, the value of R_i^{T}(a|I_{i}) will become increasingly large, which makes neural network difficult to learn the enormous increasing. This problem isn't dominated under small iterations. In practice, we can divide the cumulative regret by \sqrt{T}. In our revised paper, Eq.9 gives more discussion about the solution to this problem.  Further details please see remark 1  and Equation 9 in section 3.1
>
>
> Q5: Are optimization methods able to deal with the high variance in large IIGs?
>
> A: The high variance of the CFV is created by the large ratio between the  total number of the information set and the sampled number of the information set. Even in a small IIG game, such as Leduc 5 or 10, we can simulate the situation by sampling only a small number of information sets. For instance, we sampled only less than the 1% of the information set in our experiments, and the algorithm still converges though at a slower rate. Furthermore, variance reduction techniques such as the paper  “Variance Reduction in Monte Carlo Counterfactual Regret Minimization (VR-MCCFR) for Extensive Form Games using Baselines(2018)” can be used to further reduce the variance.
>
> Q6: Since each successive approximation is being trained from the previous NN, does this cause errors to accumulate? Q: How do approximation errors accumulate across CFR iterations?
> A:  In our current experiments, the neural network update in each iteration is already solved in approximation fashion. Yet, the algorithm still converges. In some sense, there is already an accumulation of approximation errors. However, counterfactual regret minimization framework is very robust to such errors and still leads to converge algorithm.

---

> ### Author Response · Authors · 2018-11-23
> **Reply to "requires larger experiments and the questions" (Q7-Q11)**
>
> Q7: Is minimizing MSE loss sufficient to approximate the strategy well? (since the mapping from regrets -> strategy is non-linear)
>
> A: Yes. In CFR, the behavior strategy profile for next iteration comes from the cumulative regret with regret matching algorithm. Actually, before each iteration, we don’t need to calculate behavior strategy for each information in advance. When we visit one information set, the regret matching algorithm is easy to calculate the corresponding strategy according to the exact regret in tabular CFR or the estimated regret in neural CFR.  Therefore, to inference the behavior strategy for the next iteration, we only need to estimate the cumulative regret, which is optimized by minimizing MSE loss.  In practise, we cannot directly learn the mixture strategy by cross entropy because in Eq.4  and Eq.5 the mixture strategy is weighted by \frac{\pi^{\sigma^{t}}_{i}(I_{i})}{\sum_{t=1}^{T}\pi^{\sigma^{t}}_{i}(I_{i})}, this factor is different in different iterations, and we cannot inference the original S^{T-1}(a|I_{i}) according to the normalized strategy unless we use another tabular or neural network remembers the weighting factor.
>
> Q8: I believe there is also a theoretical problem with this algorithm. In Eq. 8, they minimize the loss of CF value predictions *over the distribution of infosets in the last CFR step ("batch")*. However, this distribution may change between CFR iterations, for example if the strat_t folds 2-3 on the preflop then flop infosets with 2-3 hole cards will never be observed on iteration t+1. As a result, the NN loss will only be minimized over infosets observed in the last iteration - so the network will "forget" the regrets for all other infosets. I think this issue does not arise in these toy games because all infosets are observed at each iteration, but this is certainly not the case in real games. There are a number of ways that this issue could be addressed (e.g. train on historical infosets rather than the current batch, etc.) These would need to be explored
>
> A: When we optimize Eq.7 or Eq.8 for iteration t+1, we should initialize its parameters with the learned parameters of t-th iteration (see line3 in Algorithm 2). Without this initialization,  the neural network will forget (maybe partially forget because of the generalization) the regrets for all other infosets except the observed ones. Actually, we borrow the ideas in the field of online learning, which updates parameters by additional samples.  Even in a small IIG game, such as Leduc 5 or 10, we can simulate the situation by sampling only a small number of information sets. For instance, we sampled only less than 1% of information set in our experiments, and the algorithm still converges though at a slower rate. Furthermore, variance reduction techniques such as the paper  “Variance Reduction in Monte Carlo Counterfactual Regret Minimization (VR-MCCFR) for Extensive Form Games using Baselines(2018)” can be used to further reduce the variance.
>
> Q9:"... the original CFR only works for discrete state and action spaces...": Are the authors implying that DN-CFR addresses this limitation?
>
> A: In original CFR, the game tree is built according to  the specified action spaces (including both cards and betting money).  In the neural method, we encoding the normalized betting money (betting money/stack) as the feature of input.  Theoretically, the neural method is not limited to the integer betting money.
>
>
> Q10:"...these methods do not explicitly take into account the hidden information in a game..." Could you clarify? Is your point that these methods operate on the normal form rather than extensive form game?
> A: In NSFP, the reinforcement learning network is optimized based on the transition replay memory. When given the particular private cards of both players,  the reward in replay memory is the payoff of this game (not the expected payoff).  In CFR framework, the counterfactual value is the expected payoff over the posterior probability of the opponent’s private cards.
>
> Q11: The paragraph starting "In standard RNN..." should be reduced or moved to appendix. The exact NN architecture is not central to the ideas, and there is no experimental comparison with other architectures so we have no evidence that the architecture is relevant.
>
> A:  Honestly, the proposed neural network architecture is not the most important part of our contributions. We reorganize the structure of this section accordingly. In the experiments, network architectures and optimization methods are very important. We find a carefully designed architecture will help us achieve a faster convergence rate. Further details please see Figure 5(D).
>
> We fixed the typos in the revised paper accordingly.

---

### Official Review · AnonReviewer2 · 2018-11-03
**Appears to be a solid advance in NN applied to IIG, but I'm not an expert in this area**

**Rating:** 6
**Confidence:** 2

**Review:**

This paper proposes a pair of LSTM networks, one of which estimates the current strategy at iteration t+1 and the other estimates the average strategy after t iterations. By using these networks within a CFR framework, the authors manage to avoid huge memory requirements traditionally needed to save cumulative regret and average strategy values for all information sets across many iterations.
The neural networks are trained via a novel sampling method with lower variance/memory-requirements that outcome/external sampling, and are amendable to continual improvement by warm-starting the networks based on cloned tabular regret values.

Overall, the paper is well-written with clear definitions/explanations plus  comprehensive ablation-analyses throughout, and thus constitutes a nice addition to the recent literature on leveraging neural networks for IIG.

I did not find many flaws to point out, except I believe the paper could benefit from more extensive  comparisons in Figure 4A against other IIG methods such as Deep Stack, as well as comparing on much larger IIG settings with many more states to see how the neural CFR methods hold up in the regime where they are most needed.

Typo:  "care algorithm design" -> "careful algorithm design"

---

> ### Author Response · Authors · 2018-11-23
> **Reply to "comparing on much larger IIG settings "**
>
> Thanks for your effort in providing this detailed and useful review!
>
> We present our clarification in the following:
>
> Q: I believe the paper could benefit from more extensive  comparisons in Figure 4A against other IIG methods such as Deep Stack, as well as comparing on much larger IIG settings with many more states to see how the neural CFR methods hold up in the regime where they are most needed.
>
> A: To address this problem, we add three different kinds of experiments.
> Use small batch size, only a small subset of infosets are sampled in each iteration.  In this case, we can present the generalization ability of the neural network.
> Use small embedding size and let the number of parameters is much fewer than the number of infosets of the whole game tree. In this case, we can present the compression ability of the neural network.
> Use the larger stack to increase the size of the game tree.
> In all these three kinds of experiments, we find the neural CFR can still converge to a good strategy. Further details please see Figure 5(A), 5(B) and 5(C) in the revised paper.
>
> We fixed the typos in the revised paper accordingly.

---

> > ### Comment · AnonReviewer2 · 2018-11-26
> > **Fig 5 is nice addition, but still missing comparison in large games**
> >
> > The authors have provided a welcome new analysis in Fig. 5, in which performance in larger games was investigated (up to stack of size 15) and the compression/generalization ability of the neural net is displayed.
> >
> > While the ablation analyses and empirical investigations of the proposed method itself are quite thorough, there is still no comparison of the proposed method against a baseline neural method (e.g. something along the lines of DeepStack) on a game of realistically large size.
> >
> > I will thus keep my score the same.

---

### Official Review · AnonReviewer3 · 2018-11-08
**Isn’t it hard to learn cumulative quantities in a neural net?**

**Rating:** 5
**Confidence:** 4

**Review:**

The paper proposes a neural net implementation of counterfactual regret minimization where 2 networks are learnt, one for estimating the cumulative regret (used to derive the immediate policy) and the other one for estimating a cumulative mixture policy. In addition the authors also propose an original MC sampling strategy which generalize outcome and external sampling strategies.

The paper is interesting and easy to read. My main concern is about the feasibility of using a neural networks to learn cumulative quantities.

The problem of learning cumulative quantities in a neural net is that we need two types of samples:
- the positive examples: samples from which we train our network to predict its own value plus the new quantity,
but also:
- the negative examples: samples from which we should train the network to predict 0, or any desired initial value.

However in the approach proposed here, the negative examples are missing. So the network is not trained to predict 0 (or any initial values) for a newly encountered state. And since neural networks generalize (very well...) to states that have not been sampled yet, the network would predict an arbitrary values in states that are visited for the first time. For example the network predicting the cumulative regret may generalize to large values at newly visited states, instead of predicting a value close to 0. The resulting policy can be arbitrarily different from an exploratory (close to uniform) policy, which would be required to minimize regret from a newly visited state.  Then, even if that state is visited frequently in the future, this error in prediction will never be corrected because the target cumulative regret depends on the previous prediction. So there is no guarantee this algorithm will minimise the overall regret.
This is a well known problem for exploration (regret minimization) in reinforcement learning as well (see e.g. the work on pseudo-counts [Bellemare et al., 2016, Unifying Count-Based Exploration and Intrinsic Motivation] as one possible approach based on learning a density model).

Here, maybe a way to alleviate this problem would be to generate negative samples (where the network would be trained to predict low cumulative values) by following a different (possibly more exploratory) policy.


Other comments:
- It does not seem necessary to predict cumulative mixture policies (ASN network). One could train a mixture policy network to directly predict the current policy along trajectories generated by MC. Since the samples would be generated according to the current policy \sigma_t, any information nodes I_i would be sampled proportionally to \pi^{\sigma^t}_i(I_i), which is the same probability as in the definition of the mixture policy (4). This would remove the need to learn a cumulative quantity.
- It would help to have a discussion about how to implement (7), for example do you use a target network to keep the target value R_t+r_t fixed for several steps?
- It is not clear how the initialisation (10) is implemented. Since you assume the number of information nodes is large, you cannot minimize the l2 loss over all states. Do you assume you generate states by following some policy? Which policy?

---

> ### Author Response · Authors · 2018-11-23
> **Reply to "Isn’t it hard to learn cumulative quantities in a neural net?"**
>
> Thanks for your effort in providing this detailed and useful review!
>
> We present our clarification in the following:
>
> Q1: the feasibility of using neural networks to learn cumulative quantities:
>
> A: In each iteration, only a small subset of information sets are sampled, which may lead to the neural networks forgetting values for those unobserved information sets. To avoid such catastrophic forgetting, we used the neural network parameters from previous iterations as initialization, which gives an online learning/adaptation to the update. Furthermore, due to the generalization ability of the neural networks, even samples from a small number of information sets are used to update the new neural networks, we find that the newly updated neural networks can produce very good value for the cumulative regret and the strategy mixture. (we give related discussion in section 3.1 and add much more experimental results in Figure 5, further details please see the revised paper.)
>
> Q2: It does not seem necessary to predict cumulative mixture policies (ASN network)?
>
> A: As you say, any information nodes I_i would be sampled proportionally to \pi^{\sigma^t}_i(I_i), which is the same probability as in the definition of the mixture policy (Eq.4). Actually, if we have a large enough buffer to save all the sampled nodes, it’s easy to inference the mixture policy accordingly. However, in the large game, this large memory is expensive and impossible. Another method called reservoir sampling was used in NSFP to address a similar problem.  We borrow this idea to our method, however, the achieved mixture policy cannot converge to a low exploitability.  Actually, the third possible solution could employ the checkpoint of each current strategy, and mixture this current strategy accordingly.
>
>
> Q3: It would help to have a discussion about how to implement (7), for example do you use a target network to keep the target value R_t+r_t fixed for several steps?
>
> A: The optimization problem for the double neural networks is different from that in DQN, where the target network is fixed for several steps and only one step of gradient descent is performed. In our setting, both RSN and ASN perform several steps of gradient descent with stochastic mini-batch samples. Furthermore, in DQN, the Q-value for the greedy action is used in the update, while in our setting, we do not use greedy actions. Algorithm E gives further details on how to optimize the objectives in Equation 7 and Equation 8 (Further discussion please see the revised paper.)
>
> Q4: It is not clear how the initialisation (10) is implemented. Since you assume the number of information nodes is large, you cannot minimize the l2 loss over all states. Do you assume you generate states by following some policy? Which policy?
>
> A: Generally, Eq.10 is an idea of behavior cloning algorithm.  Clone a good initialization, and then continuously update the two neural networks using our method.  In the large extensive game, the initial strategy is obtained from an abstracted game which has a manageable number of information sets.  The abstracted game is generated by domain knowledge, such as clustering similar hand strength cards into the  same buckets. (refer to section 3.3 in the revised paper.)

---

### Public Comment · ~Marc_Lanctot1 · 2018-09-28
**Neat paper!**

I have not read it fully, but this paper looks super interesting!

Just wanted to quickly mention that you should be citing the Regression CFR paper (Waugh et al., 2015, "Solving Games with Functional Regret Estimation", AAAI) as it is clearly related work. It was the first to propose function approximation in CFR. Like one of the parts of your double network, it proposed building a regressor to predict cumulative regrets. Here is the link: https://arxiv.org/abs/1411.7974

---

> ### Author Response · Authors · 2018-09-28
> **Reply to the question: missing reference "Solving Games with Functional Regret Estimation"**
>
> Thanks for pointing out the missing reference, we will cite the paper in the revised version.

---

> ### Author Response · Authors · 2018-11-23
> **new revised version**
>
> Dear Marc,
>
> our paper is revised accordingly.  Some common questions are summarized in the comment "Paper revision 1". Further details please see the revised paper.

---

### Public Comment · ~Marc_Lanctot1 · 2018-09-28
**Lemma 2 = Lemma 1 of from MCCFR paper?**

One more thing. Is your Lemma 2 any different than Lemma 1 from Lanctot et al. 2009? If not, that fact should be cited somewhere as well (that it's a restatement of the main MCCFR lemma).

---

> ### Author Response · Authors · 2018-09-28
> **Reply to the question: Lemma 2 = Lemma 1 of from MCCFR paper?**
>
> Thanks for your suggestion. The Lemma 2 is the same with the Lemma 1 from Lanctot et al. 2009, which is cited in the background. This Lemma will be used to prove the unbiased estimation of counterfactual value for mini-batch MCCFR. We will cite Lemma 1 from Lanctot et al. 2009 in the revised version.

---

> > ### Public Comment · ~Marc_Lanctot1 · 2018-09-28
> > **Reply to Reply about Lemma 2 citation**
> >
> > Great, thanks. The citation in the background makes sense. However, I recommend you also add "(Lemma 1 of Lanctot et al. 2009)" after "Proof: " in Section E.2. As currently written, the casual reader might misinterpret the proof in E.2 as novel.

---

### Author Response · Authors · 2018-11-23
**Paper revision 1**

To answer the common questions, we provide new experimental results and revise our paper accordingly. Besides the refinement of typos and reorganization the structure of the article, this version includes the following modifications:

(1) Does neural network have generalization?
In order to inspect the generalization ability, we perform the neural CFR with small mini-batch sizes (b=50, 100, 500), where only 3.08%, 5.59%, and 13.06% information sets are observed in each iteration. According to the results in Figure 5(A), all of these settings can arrive at exploitability less than 0.1 within only 1000 iterations.

(2) Do fewer parameters still work well?
In the proposed neural architecture, the embedding size 8 and 16 will leads to 1048 and 2608 parameters respectively in no-limited leduc hold’em with size 5, both of which are much less than the tabular memory, which saves about 10^4 values. Both these two embedding sizes can converge to a good strategy profile as shown in Figure 5(B).

(3)Does neural method converge in the larger game?
Figure 5(C) presents the log-log convergence curve of different stack size 5, 10, 15. The largest game size contains over 2*10^7 states and 3.7*10^6 information sets. Let mini-batch size be 500, there are 13.06%, 2.39% and 0.53% information sets that are observed respectively in each iteration. Even though only a small subset of nodes are sampled, the double neural method can still achieve O(1/\sqrt{T}) convergence rate.

(4) Is attention in the neural architecture helpful?
Figure 5(D) presents the convergence curves of several different deep neural architectures (LSTM, LSTM plus attention, original RNN plus attention, and GRU plus attention). The recurrent neural network with LSTM cell plus attention helps us obtain a better convergence rate than the counterpart after hundreds of iterations.

---

### Author Response · Authors · 2018-11-24
**Paper revision 2**

In addition to our revision 1, which we perform the proposed double neural in a larger game containing more than 2*10^7 states with fewer parameters and sampling smaller subsets of information sets, we made an updated revision 2. In this version includes the following modification:

(1) We make more clarification about how to optimize the proposed neural network (see remark 1~3 in section 3.1) and continual improvement (see remark in section 3.3).

(2) We add a comparison for several different neural architectures (see Figure5(D)), in the experiments, sequential recurrent neural network converges faster than the fully connected neural network. The architecture of LSTM plus attention helps us obtain the fastest converging strategy profile.

---

### Meta-Review · Area_Chair1 · 2018-12-13
**Interesting ideas with better motivation needed for soundness**

**Confidence:** 4
**Recommendation:** Reject

**Metareview:**

The reviewers agreed that there are some promising ideas in this work, and useful empirical analysis to motivate the approach. The main concern is in the soundness of the approach (for example, comments about cumulative learning and negative samples). The authors provided some justification about using previous networks as initialization, but this is an insufficient discussion to understand the soundness of the strategy. The paper should better discuss this more, even if it is not possible to provide theory. The paper could also be improved with the addition of a baseline (though not necessarily something like DeepStack, which is not publicly available and potentially onerous to reimplement).